# Quantitative perturbation-phenotype maps reveal nonlinear responses underlying robustness of PAR-dependent asymmetric cell division

Nelio T. L. Rodrigues[1], Tom Bland[1,2], KangBo Ng[1,2], Nisha Hirani[1], Nathan W. Goehring[1,2] *

1 The Francis Crick Institute, London, United Kingdom, 2 Institute for the Physics of Living Systems, University College London, London, United Kingdom

* nate.goehring@crick.ac.uk

## Abstract

A key challenge in the development of an organism is to maintain robust phenotypic outcomes in the face of perturbation. Yet, it is often unclear how such robust outcomes are encoded by developmental networks. Here, we use the *Caenorhabditis elegans* zygote as a model to understand sources of developmental robustness during PAR polarity-dependent asymmetric cell division. By quantitatively linking alterations in protein dosage to phenotype in individual embryos, we show that spatial information in the zygote is read out in a highly nonlinear fashion and, as a result, phenotypes are highly canalized against substantial variation in input signals. Our data point towards robustness of the conserved PAR polarity network that renders polarity axis specification resistant to variations in both the strength of upstream symmetry-breaking cues and PAR protein dosage. Analogously, downstream pathways involved in cell size and fate asymmetry are robust to dosage-dependent changes in the local concentrations of PAR proteins, implying nontrivial complexity in translating PAR concentration profiles into pathway outputs. We propose that these nonlinear signal-response dynamics between symmetry-breaking, PAR polarity, and asymmetric division modules effectively insulate each individual module from variation arising in others. This decoupling helps maintain the embryo along the correct developmental trajectory, thereby ensuring that asymmetric division is robust to perturbation. Such modular organization of developmental networks is likely to be a general mechanism to achieve robust developmental outcomes.

## Introduction

Developmental systems possess a remarkable ability to maintain stable phenotypes in the face of perturbations including variable gene expression, noise, environmental conditions, physical insult or constraints, or even mutational load. This has led to the notion that systems have

**Data Availability Statement:** All underlying numerical data and relevant analysis code are

available at https://doi.org/10.25418/crick.
27153459.

**Funding:** This work was supported by the Francis
Crick Institute (to N.W.G), which receives its core
funding from Cancer Research UK (CC2119 to N.
W.G.), the UK Medical Research Council (CC2119
to N.W.G.), and the Wellcome Trust (CC2119 to N.
W.G.). The funders had no role in study design,
data collection and analysis, decision to publish, or
preparation of the manuscript.

**Competing interests:** The authors have declared
that no competing interests exist.

**Abbreviations:** ASI, asymmetry index; NEBD,
nuclear envelope breakdown; NGM, nematode
growth media; ROI, region of interest.

evolved to minimize variance in outputs, i.e., phenotypic traits, in the face of perturbations or
variation in input signals, rendering them robust [1,2].

The robustness of phenotypic outcomes has many origins and includes mechanisms that
act at multiple scales of organization [3–5]. Molecular buffering or dosage compensation
mechanisms can directly compensate for variance in network components [6–9]. Network fea-
tures, such as activity-dependent feedback, saturation, or kinetic linkage, can also ensure that
input–output functions of the network remain robust to variation in particular components
[10–14]. Finally, the organism may have network-extrinsic mechanisms that allow systems to
correct for variability in network outputs [15–17].

Regardless of its ultimate mechanistic origin in a given system, robustness is typically asso-
ciated with nonlinear signal-response curves. Such nonlinearities yield threshold-like behav-
iors that effectively canalize variable input parameters into similar developmental trajectories,
thereby allowing them to converge upon similar outcomes [18,19]. In the case of mutational or
allelic variation, such mechanisms can yield highly nonlinear genotype-phenotype maps asso-
ciated with phenotypic canalization [5].

In *Caenorhabditis elegans* and related species, the first cell division is nearly always asym-
metric in both size and fate, the latter manifest as cell cycle asynchrony between daughter cells
and ultimately their divergence into distinct lineages. Although the precise magnitude of these
asymmetries can vary between species, both cell size asymmetry and cell cycle asynchrony are
highly reproducible within a given species [20–23]. Moreover, at least in *C. elegans*, asymmet-
ric division is robust to genetic and environmental perturbations, including both temperature
variation and physical deformation [24–33]. While embryos can tolerate some variation in cell
size asymmetry and cell cycle asynchrony, in part due to compensatory behaviors that occur
later in development [34–36], the design principles that underlie this robustness of division
asymmetry itself remain largely unknown.

Asymmetric division of the zygote is under direct control of a set of conserved cell polarity
proteins known as the PAR(-*titioning defective*) proteins [37]. The PAR proteins consist of 2
antagonistic groups of membrane-associated proteins that segregate into opposing anterior
and posterior membrane domains during the first division [38,39]. Segregation is triggered by
a set of semi-redundant symmetry-breaking cues that induce initial asymmetries in the distri-
bution of PAR proteins in the zygote. These asymmetries are then reinforced and maintained
through a core set of feedback interactions to generate robustly segregated anterior and poste-
rior PAR domains. Once formed, these PAR domains direct the spatial organization of down-
stream processes that orchestrate the size and fate asymmetry of cell division [40]
(summarized in Fig 1A).

The core feedback circuits underlying segregation of PAR proteins in the zygote are a set of
mutually antagonistic (double negative) interactions. Specifically, anterior PAR proteins
(aPARs)—PAR-3, PAR-6, PKC-3, and CDC-42—restrict membrane association of the oppos-
ing posterior (pPAR) proteins—PAR-1, PAR-2, LGL-1, and CHIN-1—through their phos-
phorylation by PKC-3 [41–44]. Conversely, pPAR proteins exclude aPARs through the
phosphorylation of PAR-3 by PAR-1 [30,45] and inhibition of active CDC-42 by the CDC-42
GAP CHIN-1 [46–48]. This reciprocal negative feedback is thought to be complemented by
within-group positive feedback, e.g., cooperative membrane binding [49–52].

Due to the mutually antagonistic nature of feedback, PAR polarity generally relies on bal-
ancing aPARs and pPARs, which act via their respective polarity kinases PKC-3 and PAR-1
(reviewed in [29,38,39]). Mathematical models based on mutual antagonism predict sensitivity
to dosage changes, which manifest as shifts in membrane concentrations, relative domain sizes,
and ultimately collapse of polarity depending on the magnitude of the perturbation, behaviors
which have been generally confirmed in vivo, at least at a qualitative level [25,53–55]. Mutations

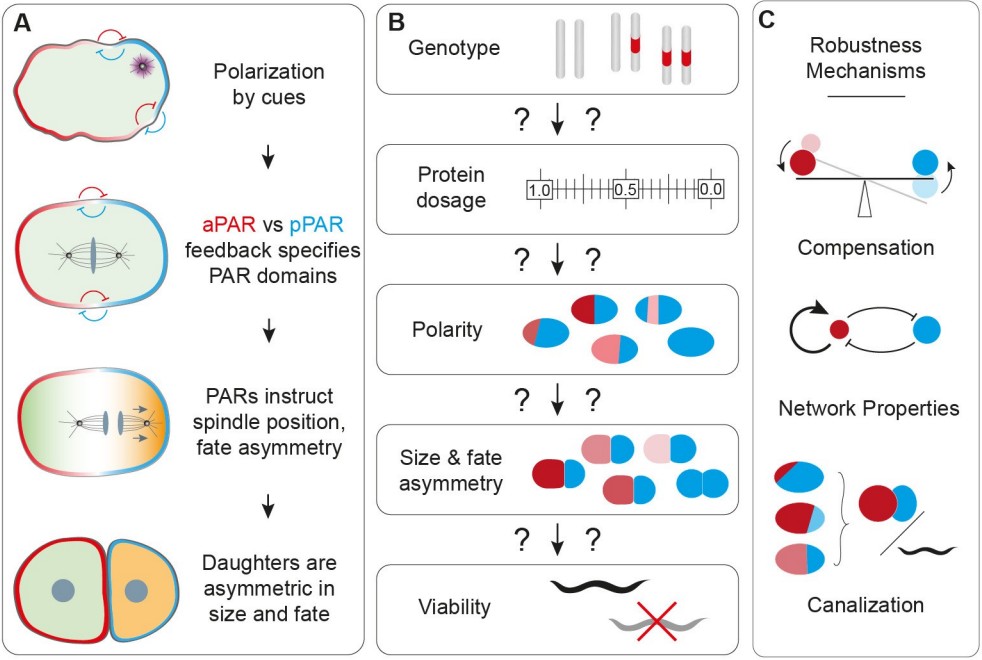

**Fig 1. Potential mechanisms for dosage robustness in asymmetric division. (A)** Schematic of the asymmetric division pathway in *C. elegans* zygotes. A local cue induces asymmetry of PAR proteins which is then reinforced by mutual antagonism between anterior and posterior PAR proteins (aPAR, pPAR) to generate stable domains. PAR proteins then spatially regulate downstream processes to drive division asymmetry. Due to this mutually antagonistic relationship, aPAR and pPAR protein levels/activities must be balanced to achieve proper polarity. **(B)** While we know that polarity, asymmetric division, and viability rely on PAR proteins and that animals heterozygous for *par* mutations are generally viable, the quantitative relationships between genotype, protein dosage, polarity, asymmetric division, and viability have not been measured, leaving the root mechanisms underlying robustness of division asymmetry unclear. **(C)** Mechanisms underlying robustness: (1) compensation—PAR protein levels actively adapt to gene/protein dosage changes to restore balance; (2) network properties—features of the network, such as feedback circuits, compensate for dosage imbalance to maintain stable polarity signals; (3) canalization—downstream asymmetric division pathways that drive size/fate asymmetry are robust to variability in polarity signals.

in pPAR genes can be genetically suppressed by partial depletion of aPARs and vice versa. Polarity outcomes are also sensitive to ectopic overexpression of individual PAR proteins, particularly in sensitized backgrounds [27,33,55–57]. At the same time, there is a notable lack of developmental phenotypes in embryos heterozygous for mutations in *par* genes or indeed in the vast majority of genes essential for early embryogenesis [37,57,58].

Thus, there is an apparent disconnect between the predicted reliance of PAR polarity on balancing aPAR and pPAR activity on one hand, and the apparent robustness of asymmetric division to gene/protein dosage variation on the other. This disconnect led us to quantitatively examine the coupling between symmetry-breaking cues, PAR polarity, and asymmetries in size and fate, with a specific focus on understanding the impact of perturbation in PAR protein/gene dosage (Fig 1B). By combining established methods for manipulation of protein dosage in *C. elegans* [59] with recently developed image quantitation-based workflows [50,60], we directly relate dosage to phenotype in individual embryos. Our data support a model in which pathway responses are canalized against variation in spatial signals at multiple levels, leading to decoupling between symmetry-breaking, polarity, spindle positioning, and fate segregation modules. This decoupling effectively insulates individual modules from variability arising elsewhere in the pathway, helping to ensure reproducible outcomes in both size and fate asymmetry during asymmetric division.

## Results

We reasoned that robustness of asymmetric division in the *C. elegans zygote* to changes in *par* gene or PAR protein dosage could arise from a variety of mechanisms known to contribute to the robustness of developmental processes: (1) Dosage compensation: animals harboring a loss of function *par* allele could up-regulate expression of the remaining functional allele, ensuring normal concentrations. Alternatively, compensatory changes to levels of other PAR proteins within the network could act to restore normal function, as in [6]. (2) PAR network properties: features of the PAR network, such as feedback circuit design, render its outputs insensitive to modest changes in dosage of any given component, as in [13,14]. (3) Canalization: the downstream asymmetric division machinery is insensitive to dosage-dependent variation in PAR concentration profiles/domain size, as in [19]. In other words, PAR distributions may be dosage sensitive, but downstream pathways are robust to these changes (Fig 1C).

### Compensatory dosage regulation cannot explain robustness to heterozygosity in *par* genes

We first asked whether embryos exhibited dosage compensation. Chromosome-wide dosage compensation is well known in the context of sex chromosomes: Gene expression is systematically up- or down-regulated to account for differences in sex chromosome number in males and females [61]. However, dosage compensation of individual autosomal genes is less well understood. Systematic transcriptional analysis suggests that the degree of compensation can vary substantially at the level of individual genes, though the vast majority show no or only partial compensation [62,63].

We initially looked for evidence of dosage compensation in animals heterozygous for mutant alleles of 2 polarity genes *par-6* and *par-2*, as representatives of aPAR and pPAR genes, respectively. Due to maternal provision to oocytes, the mRNA and protein composition is primarily determined by the mother's genotype. Thus, for simplicity, hereafter we refer to embryos by the genotype of the mother, i.e., heterozygous embryos = embryos from heterozygous mothers. To test whether compensation occurs, we applied spectral autofluorescence correction using SAIBR [60] to accurately quantify and compare GFP levels in embryos of 3 genotypes: (1) homozygous for endogenously *gfp*-tagged alleles (*gfp/gfp*) in which all protein is GFP-tagged; (2) heterozygous embryos carrying a single tagged allele together with an untagged wild-type allele (*gfp/+*), in which we expect GFP-labeled protein to constitute roughly half of total protein; and (3) heterozygous embryos carrying a single tagged allele over either a null allele or an allele that can be selectively depleted by RNAi (*gfp/-*). For perfect dosage compensation, we would expect levels of GFP in *gfp/gfp* and *gfp/-* to be similar (Fig 2A). However, we find that embryos from *gfp/-* worms expressed levels of GFP that were only modestly increased relative to *gfp/+*, and well below those of *gfp/gfp* embryos, suggesting only partial up-regulation (Figs 2B–2D and S1). Similar results were obtained for other *par* genes examined, including *par-1*, *par-3*, and *pkc-3*, which showed modest to no compensation in heterozygotes (Figs 2D and S1).

Because of the requirement for balanced activity of aPAR and pPAR proteins [38,39], we also asked whether down-regulation of other components in the PAR network could help explain the robustness of embryos to dosage changes in individual PAR proteins. In other words, would depletion of a given PAR protein lead to reduction in the concentration of opposing PAR proteins? We therefore performed progressive depletion of either PAR-2 or PAR-6 by RNAi and monitored the dosage of the other. We found that dosage of PAR-2 remained constant across the full range of PAR-6 depletion conditions and that PAR-6 dosage

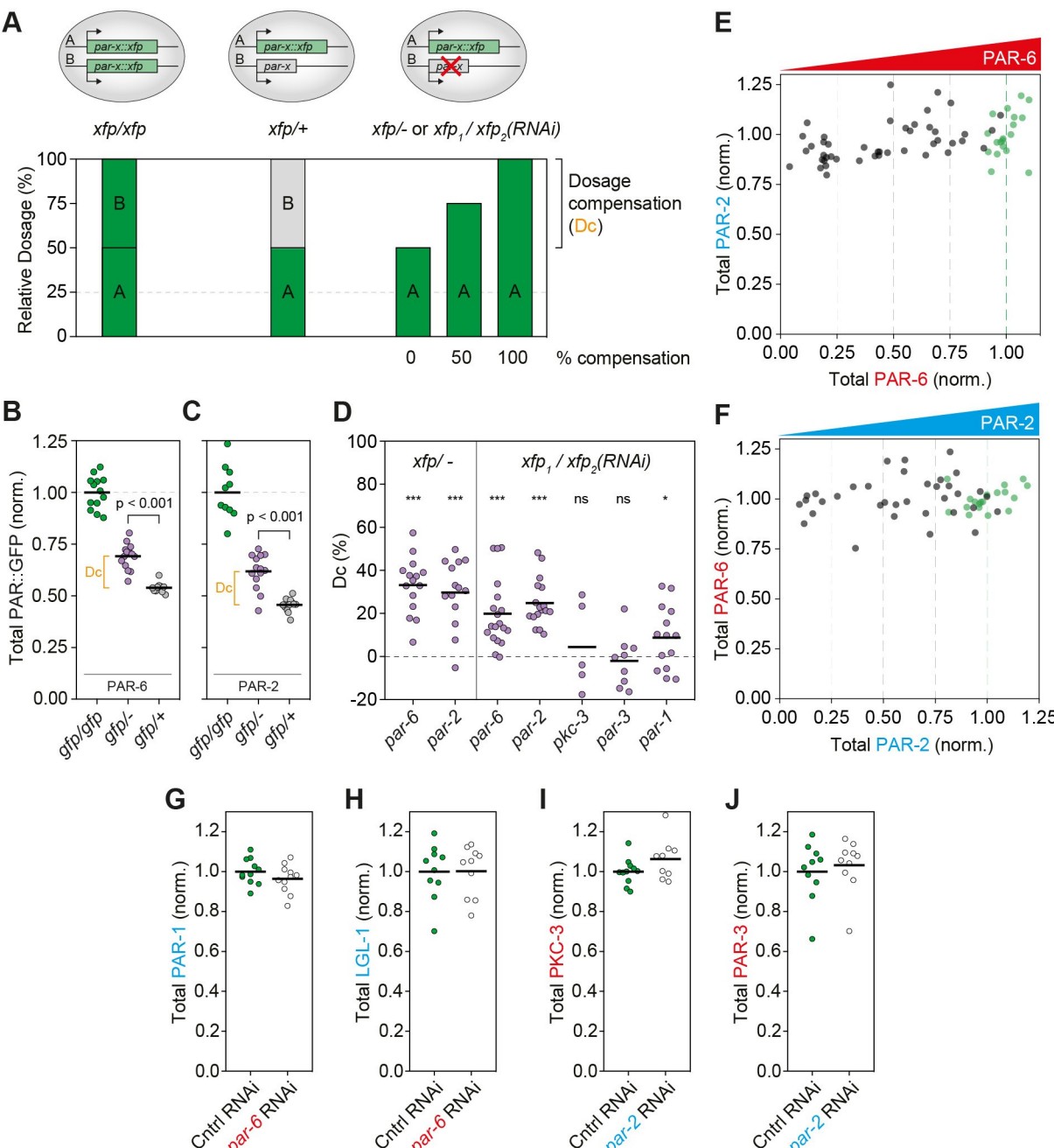

**Fig 2. Minimal compensatory regulation in response to *par* gene/protein dosage changes. (A)** Schematic for dosage compensation assay. Levels of XFP (GFP or mNG) were measured for embryos of 3 genotypes: homozygous, carrying 2 copies of an *XFP::par* allele (*xfp/xfp*); heterozygous, carrying 1 copy of *XFP::par* allele and 1 untagged allele (*xfp/+*), which is expected to express XFP at approximately 50% levels of homozygotes; and heterozygous, carrying 1 copy of the *XFP::par* allele and either a mutant or RNAi-silenced allele (*xfp/-* or *xfp/RNAi*). Dosage compensation is quantified as the degree of excess XFP signal in *xfp/-* or *xfp/RNAi* embryos, expressed as the fraction of the difference in XFP signal between *xfp/xfp* and *xfp/+* animals. **(B, C)** Normalized GFP concentrations of PAR-6::GFP (B) or GFP::PAR-2 (C) as measured in embryos with the indicated genotypes: homozygous (*gfp/gfp*), heterozygous mutant (*gfp/-*), and heterozygous untagged (*gfp/+*) genotypes. Unpaired *t* test. **(D)** Modest or no dosage compensation exhibited for various *par::XFP* gene fusions when expressed in a heterozygous condition together with either a mutant (*xfp/-*) or an RNAi-silenced allele (*xfp₁/xfp₂(RNAi)*). ***$p < 0.0001$, *$p < 0.05$, one-sample *t* test. Additional details for allele-specific RNAi in S1 Fig. **(E)** Total PAR-2 concentration is constant as a function of PAR-6 dosage. Embryos expressing both mCh::PAR-2 and PAR-6::mNG from the endogenous loci were subjected to progressive depletion of PAR-6 by RNAi and total concentrations of mNG and mCh measured. Green data points are embryos treated with control RNAi (i.e., showing wild-type protein levels). **(F)** Total PAR-6 concentration is constant as a function of PAR-2 dosage. Fluorescence tags as in (E), but embryos were subjected to progressive depletion of PAR-2 by RNAi. **(G, H)** PAR-1 and LGL-1 concentrations are unchanged in *par-6(RNAi)*. **(I, J)** PKC-3 and PAR-3 concentrations are unchanged in *par-2(RNAi)*. In B–D, G–J, individual embryo values shown with mean indicated. The raw data underlying this figure can be found at https://doi.org/10.25418/crick.27153459.

was similarly constant across the full range of PAR-2 depletion (Fig 2E and 2F). Consistent with these results, we found that the other posterior PAR proteins PAR-1 and LGL-1 were unchanged in PAR-6-depleted animals (Fig 2G and 2H), while the levels of anterior PAR proteins PKC-3 and PAR-3 were unchanged in PAR-2-depleted animals (Fig 2I and 2J). Thus, there do not appear to be coordinated alterations in protein amounts to compensate for changes in the dosage of a given PAR protein.

We conclude that *C. elegans* embryos do not exert homeostatic regulation of PAR concentrations in response to dosage changes. It is possible that modest up-regulation of protein amounts for some *par* genes (*par-1*, *par-2*, *par-6*) could partially contribute to stable phenotypes in heterozygotes. However, embryos heterozygous for *par-3* and *pkc-3* did not exhibit such increases (Fig 2D), suggesting that partial up-regulation is neither a general adaptation of *par* genes nor a requirement for the reported viability of *par* heterozygotes. Moreover, the limited degree of up-regulation where it exists means that heterozygotes are viable despite harboring 30% to 50% less PAR protein than wild type, raising the question of how dosage variation impacts signaling activity, polarity, and ultimately asymmetric division.

## Asymmetric division is robust to changes in PAR dosage

We next asked whether asymmetric division phenotypes are, in fact, robust to variation in PAR dosage. We focused on 2 key outputs: daughter size asymmetry, which is controlled by asymmetric spindle positioning, and cell cycle asynchrony of the AB and P1 daughters, which is manifest as a roughly 2-min cell cycle delay in division timing and is a commonly used proxy for the asymmetric partitioning of cytoplasmic fate determinants (schematic in Fig 3A and 3B) [22,37,64,65].

To broadly determine how asymmetry depends on PAR protein dosage, we scored relative size asymmetry and division time asynchrony in embryos heterozygous for *par-1*, *par-2*, *par-3*, and *par-6* (Fig 3A and 3B). In all heterozygotes, we observed only minor statistically nonsignificant changes in size asymmetry and division asynchrony that were within the standard deviation observed in wild-type embryos. Thus, heterozygotes exhibit robust control of both size and fate asymmetry.

We then used progressive depletion of PAR proteins by RNAi to quantify the relationship between protein dosage and division asymmetry. Plots of dosage versus timing asynchrony for embryos depleted for PAR-1, PAR-2, or PAR-6 were generally nonlinear, with inflection points located at or near the point of 50% depletion (Fig 3C–3E). Similarly, for all 3 proteins analyzed, (PAR-1, PAR-2, and PAR-6) size asymmetry remained within the wild-type range until depletion approached 50%. As expected, as embryos approached these inflection points, we observed a peak in phenotypic variance. In the case of PAR-2 and PAR-6 (Fig 3F and 3G), asymmetry then rapidly declined as depletion was extended beyond 50%. By contrast, depletion of PAR-1 beyond 50% yielded less striking changes (Fig 3H). A shoulder is still evident as dosage levels cross the ~50% level, but asymmetry declined only weakly thereafter (see Discussion).

Overall, we find that division asymmetry is robust to variation in PAR protein dosage, with asymmetric division phenotypes remaining at or near wild type for depletion of individual PAR proteins by up to approximately 50%.

## Overall polarity is robust to changes in PAR dosage despite variation in local PAR concentrations

To achieve asymmetric division, the PAR proteins must provide the appropriate spatial signals to downstream pathways. Given the robustness of asymmetric division phenotypes to PAR

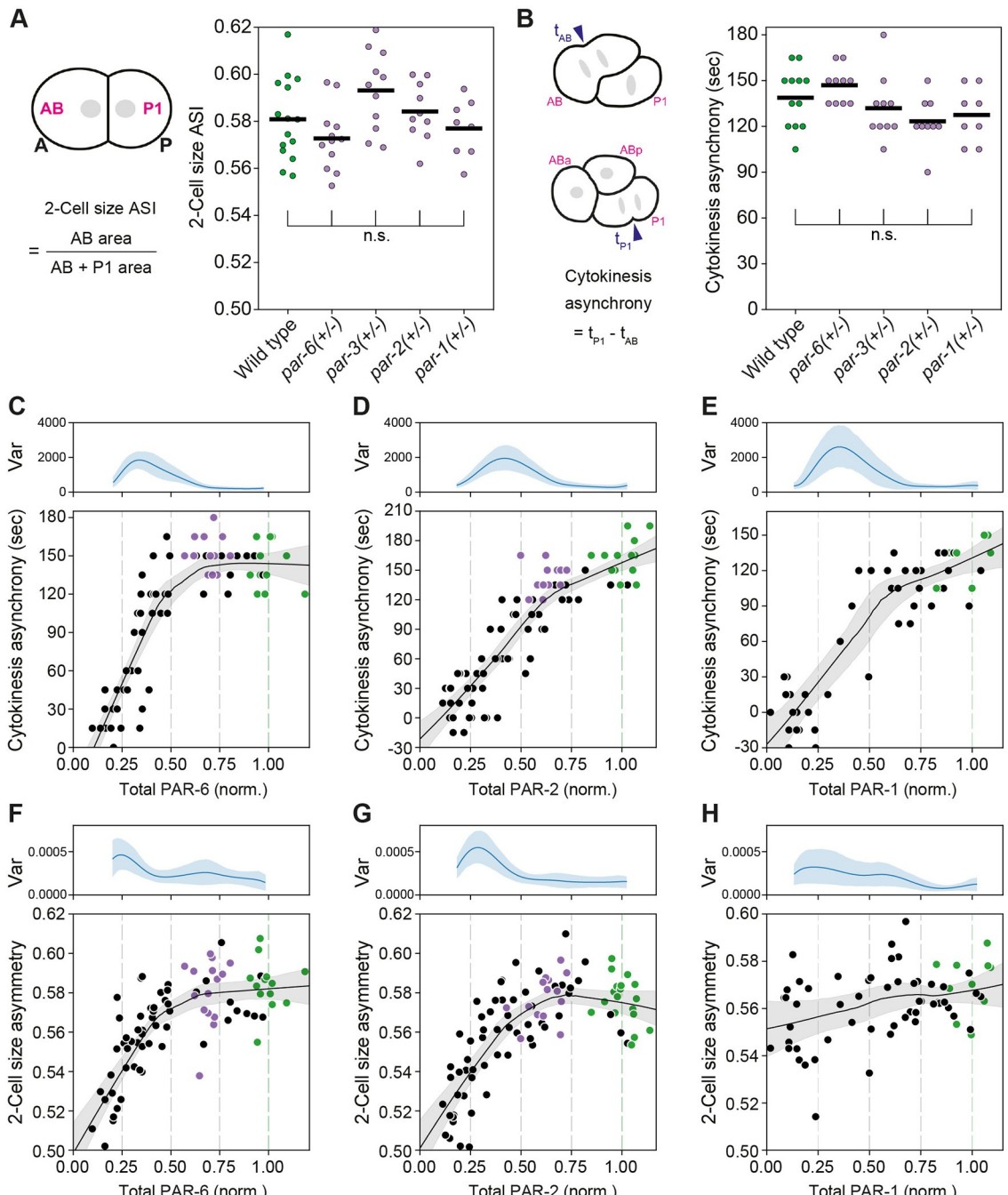

**Fig 3. Division asymmetry is robust to changes in PAR dosage. (A, B)** Size asymmetry (A) and asynchrony in cleavage furrow initiation (B) for AB and P1 blastomeres of 2-cell embryos heterozygous for mutations in the *par* genes indicated. Genotypes: +/+ (wild type), *par-6(tm1425/+)*, *par-3(tm2716/+)*, *par-2(ok1723/+)*, and *par-1(tm2524/+)*. One-way ANOVA (vs. wild type), Dunnett's correction. **(C–H)** AB vs. P1 asynchrony (C–E) and size asymmetry (F–H) as a function of total dosage of PAR-6 (C, F), PAR-2 (D, G), and PAR-1 (E, H). Data for individual embryos subject to RNAi shown (black), compared with embryos from wild-type control (green) and in the case of PAR-2 and PAR-6, heterozygous animals (*gfp/-*, purple). Lines indicate LOWESS smoothing fit with 95% confidence interval determined by bootstrapping to help visualize trends. Phenotypic variance (Var, see Methods) as a function of dosage is indicated above each panel. The raw data underlying this figure can be found at https://doi.org/10.25418/crick.27153459.

dosage, we were curious how dosage impacted PAR distributions in the embryo. Theoretical models based on mutual antagonism generally predict that PAR patterns should be dosage sensitive. Experiments have further shown that over- and under-expression can lead to changes in boundary position, ectopic polarity domains, or loss of polarity [51,53,55]. However, we lack rigorous measurements to quantitatively link dosage changes to alterations in PAR protein patterns.

To this end, we quantified the distribution of PAR-2 and PAR-6 in embryos subject to progressive depletion of one or the other protein by RNAi and quantified changes in local membrane concentration and PAR asymmetry, in this case derived from the signal weighted contributions of both proteins to provide a measure of overall polarity (asymmetry index, ASI —see Methods for calculation). We focused on embryos near the time of nuclear envelope breakdown (NEBD) when the effects of polarity cues are reduced and polarity is actively maintained by cross-talk between aPAR and pPAR proteins [25,66].

We found that progressive reduction of either PAR-6 or PAR-2 dosage was accompanied by a steady reduction in membrane concentration within their respective domains (Figs 4A–4C, 5A–5C, S2A, and S2B). We obtained similar results for PAR-1, PAR-3, and PKC-3, with heterozygous embryos exhibiting roughly 50% reductions in concentrations at the membrane (S2C–S2E Fig). Thus, there does not appear to be any mechanism to stabilize membrane concentrations in the face of changing dosage as occurs in wave pinning-like models for polarity where changes in boundary position can, at least to some degree, buffer the effects of dosage changes on membrane concentrations [53,67].

This progressive reduction of PAR-6 and PAR-2 also impacted the resulting patterns of PAR protein localization, particularly as relevant dosage was reduced below 50%. The effects were most striking for PAR-6 depletion. Here, modest reductions in PAR-6 amounts led to invasion of the anterior pole by PAR-2 (Fig 4A, 4B and 4D). Such anterior domains are thought to reflect the response of PAR-2 to secondary cues that are normally suppressed in wild-type embryos [26,68]. These ectopic domains became increasingly frequent as depletion approached 50% (Fig 4A, 4B and 4D). Curiously, for intermediate levels of depletion of between 25% and 75%, the behavior of the network was bimodal (orange highlights; Figs 4D, 4E, and S2). For similar levels of depletion, 1 population of embryos maintained effectively wild-type levels of PAR asymmetry (ASI > ~0.9). The second exhibited reduced asymmetry (ASI < 0.75) which correlated with PAR-6 concentrations, suggesting a direct relationship between aPAR activity and the relative amounts of PAR-2 at the 2 poles in this regime. This switch between regimes seems consistent with a minimum threshold level of anterior aPAR activity required to reliably suppress PAR-2 domain induction by cryptic anterior cues. Such a threshold would help ensure that most embryos maintain normal levels of PAR asymmetry (ASI > 0.9) so long as PAR-6 dosage remains above 50% (Fig 4E).

For PAR-2 depletion, the effects were less obvious. While PAR-2 domain size progressively declined, becoming undetectable as dosage was reduced below ~25%, PAR-6 domains remained relatively stable (Fig 5A and 5B). While membrane concentrations declined, accompanied by modest expansion of PAR-6 into the posterior (Fig 5D), PAR-6 never became completely uniform. Thus, the composite, signal-weighted ASI remained high across the full range of PAR-2 dosage (Fig 5E). This stability of aPARs is ostensibly in disagreement with coarse grained 2-component reaction-diffusion models that rely on competitive aPAR and pPAR domains to maintain polarity (e.g., [25,51,53,54]). However, more recent work has identified additional stabilizing features that limit posterior spread of aPARs, including other pPARs such as CHIN-1 that act during the polarity maintenance phase, PAR-dependent cortical flows that stabilize PAR domain boundaries, and positive feedback among aPAR proteins [47,48,52,69].

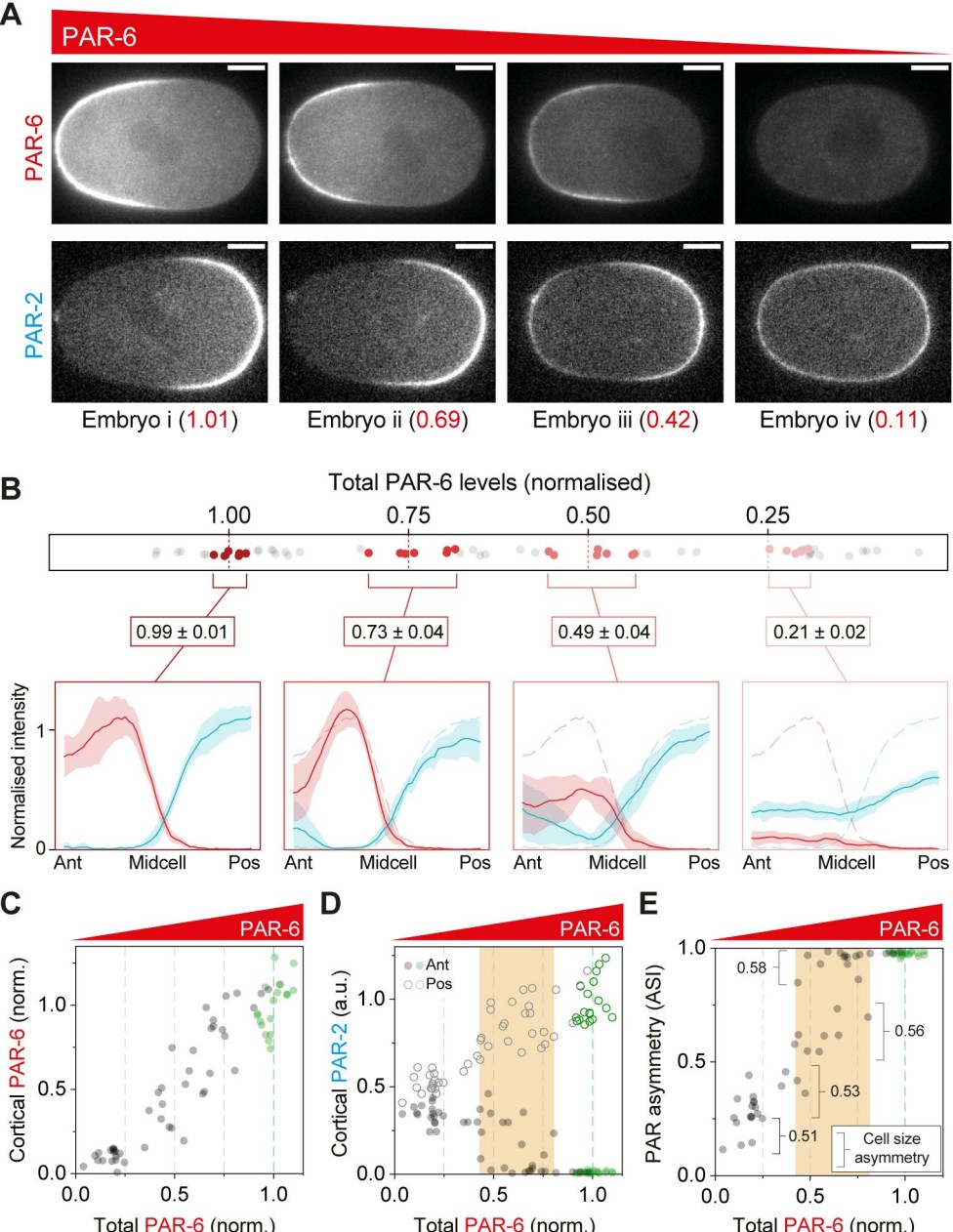

**Fig 4. Robustness of polarity to PAR-6 dosage changes. (A, B)** Evolution of PAR-2 and PAR-6 profiles as a function of the dosage of PAR-6 (A, B). Embryos expressing PAR-6::mNG and mCh::PAR-2 (NWG0268) were subject to progressive depletion of PAR-6 by RNAi and dosage measured relative to mean control levels. (A) Sample embryos shown with the dosage of the relevant PAR protein indicated. (B) To illustrate changes in concentration profiles, 7 embryos closest to the indicated dosage levels (1.0, 0.75, 0.5, and 0.25) were selected, membrane concentration profiles extracted and averaged. Mean ± SD shown. Dashed lines in 0.75, 0.5, and 0.25 dosage profiles are the mean profiles for dosage = 1.0 for comparison. **(C)** Membrane concentrations of PAR-6 decline with total PAR-6 dosage. **(D)** Reduction of PAR-6 allows invasion of PAR-2 at the anterior pole. Note appearance of anterior PAR-2 (closed circles) as PAR-6 dosage approaches 0.5. **(E)** The relationship between PAR asymmetry (ASI) and PAR-6 dosage is bimodal. For dosage >~0.8 all embryos exhibit normal asymmetry. As PAR-6 levels drop below 0.75, there is a population of embryos that retain normal asymmetry (ASI > 0.9), but a second population appears in which PAR asymmetry is reduced (ASI < 0.75) and varies linearly with PAR-6 dosage, which generally corresponds to embryos with substantial anterior PAR-2 localization. Below dosage of approximately 0.5, the population exhibiting normal asymmetry disappears. Numbers indicate mean division size asymmetry for embryos within the given ASI bins. Green data points in (C–E) are control RNAi. Orange region in (D, E) highlight range of PAR-6 dosages exhibiting bimodal phenotypes. Note that ASI in (E) is a signal-weighted composite of PAR-2 and PAR-6 asymmetry in individual embryos (see Methods). Scale bars, 10 μm. The raw data underlying this figure can be found at https://doi.org/10.25418/crick.27153459.

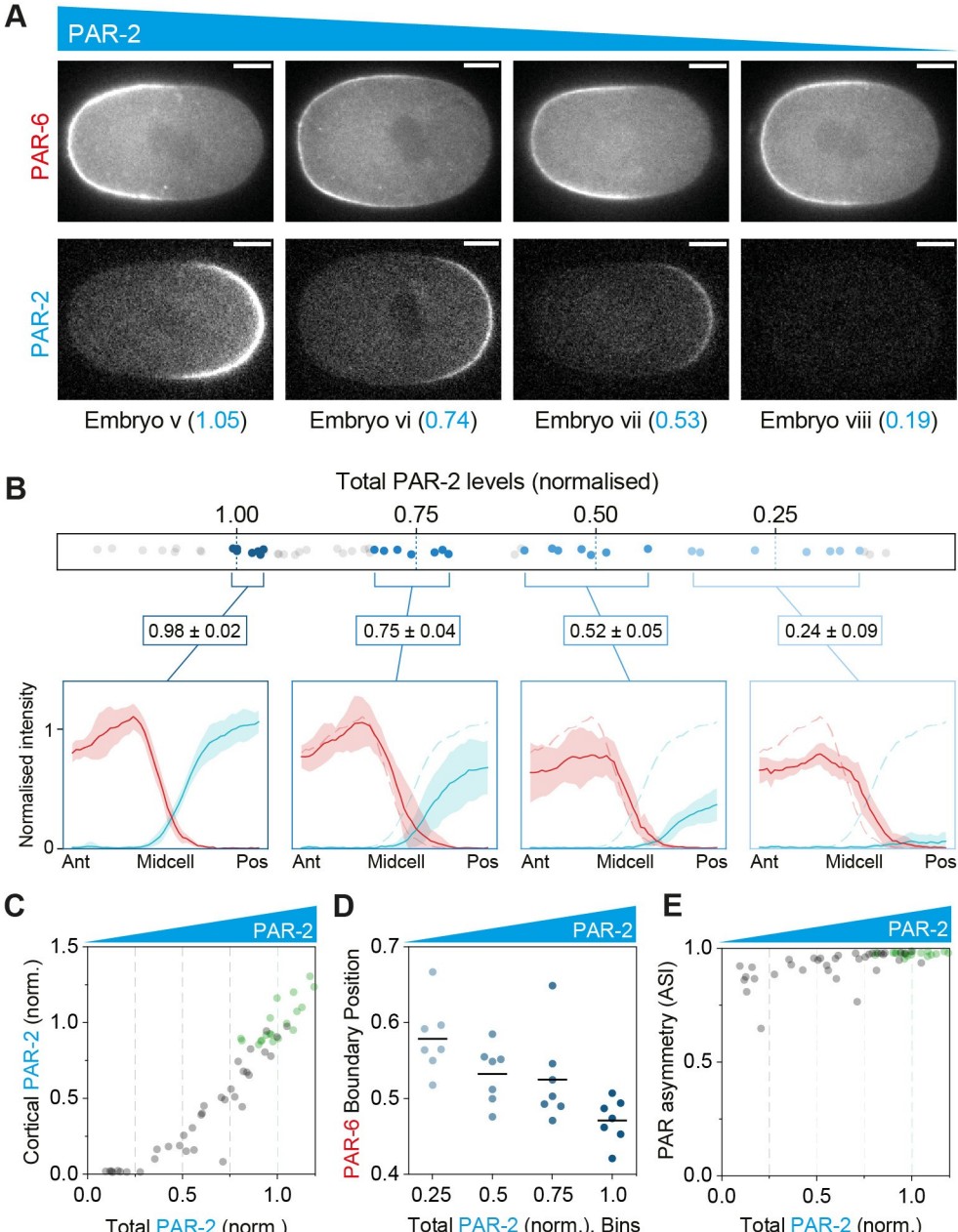

**Fig 5. Robustness of polarity to PAR-2 dosage changes. (A, B)** Evolution of PAR-2 and PAR-6 profiles as a function of the dosage of PAR-2 (A, B). Embryos expressing PAR-6::mNG and mCh::PAR-2 (NWG0268) were subject to progressive depletion of PAR-2 by RNAi and dosage measured relative to mean control levels. (A) Sample embryos shown with the dosage of the relevant PAR protein indicated. (B) To illustrate changes in concentration profiles, 7 embryos closest to the indicated dosage levels (1.0, 0.75, 0.5, and 0.25) were selected, membrane concentration profiles extracted and averaged. Mean ± SD shown. Dashed lines in 0.75, 0.5, and 0.25 dosage profiles are the mean profiles for dosage = 1.0 for comparison. **(C)** Membrane concentrations of PAR-2 decline linearly with total PAR-2 dosage. **(D)** PAR-6 domain size (boundary position relative to anterior pole) as a function of PAR-2 dosage for the PAR-2 dosage bins in (B). Individual data points and mean shown. $p$ = 0.0002, ANOVA, test for trend. **(E)** Overall PAR asymmetry (ASI) is only weakly affected by PAR-2 reductions due to the stability of aPAR domains. Note that ASI in (E) is a signal-weighted composite of PAR-2 and PAR-6 asymmetry in individual embryos (see Methods). Scale bars, 10 μm. The raw data underlying this figure can be found at https://doi.org/10.25418/crick.27153459.

Taken together, our data suggest that overall PAR asymmetry (at least as reflected in the ASI) is generally robust to relative depletion of PAR proteins by up to ~50%. Such robustness of asymmetry is likely part of the answer as to why asymmetric division phenotypes are robust to dosage changes. At the same time, dosage reductions are associated with progressive changes in other quantitative features of PAR protein localization, such as concentration profiles, peak membrane concentrations, domain boundaries, and levels of PAR proteins in the "wrong" domain. Thus, the PAR network is clearly sensitive to alterations in PAR protein dosage, even within the regime in which overall polarity is maintained and division asymmetry is normal. Finally, we noted that asymmetric division outcomes did not correlate well with our measures of overall PAR asymmetry in embryos. For example, in PAR-6 rundowns, 2-cell size asymmetry was initially stable to changes in ASI, only exhibiting an abrupt collapse as ASI declined to approximately 0.5 or less (S2H Fig). In PAR-2 rundowns, we also saw abrupt loss of both size and timing asymmetry as PAR-2 dosage declined to 0.5 and below (Fig 3D and 3G) despite ASI remaining normal across nearly the full range of PAR-2 dosage, mostly because PAR-6 remains clearly polarized (Fig 5E). These observations therefore raise questions regarding the precise signals provided by the PAR network to downstream pathways and how they are interpreted to ensure robust outcomes in the face of quantitative changes in PAR outputs, a topic we address in the next section.

### Asymmetric division pathways canalize variation in cortical PAR input signals

Asymmetric division pathways are thought to be controlled by local activity of PAR proteins, particularly those of the key polarity kinases PAR-1 and PKC-3 [70–72]. Yet as we have shown, the phenotypic endpoints of asymmetric division remain wild type despite significant changes in local PAR concentrations. How can we square these observations? Despite substantial insight into the molecular mechanisms that underlie asymmetric division, we have little quantitative data on how polarity is interpreted by downstream pathways. In particular, to what degree do downstream pathways "see" changes in local PAR concentrations or are they only sensitive to large scale changes in polarity? We therefore sought to characterize intermediate functional readouts of the pathways underlying size and fate asymmetry.

**Size asymmetry.** Size asymmetry of P0 daughters arises from posterior spindle displacement during late metaphase/anaphase of P0 mitosis [40]. Posterior spindle displacement shifts the division plane towards the posterior by a characteristic distance, thereby creating a smaller P1 and larger AB cell. Displacement is induced by PKC-3-dependent asymmetries in the number and/or activity of cortical force generators consisting of dynein, LIN-5(NuMa), GPR-1/2 (LGN), and Gαi, which exert a pulling force on astral microtubules that reach the cortex [70,73–79].

To probe sensitivity of the spindle positioning pathway, we used embryos heterozygous for mutations in *par-2* or *par-6* to achieve intermediate dosage reductions in a regime that does not impact size asymmetry (see Figs 2 and 3) and quantified key readouts of the spindle positioning pathway by monitoring spindle pole position. This included spindle elongation, spindle displacement along the A-P axis, and transverse spindle oscillations. Transverse oscillations arise during spindle elongation and are thought to depend on a critical threshold pulling force, and thus serve as a sensitive readout of changes in the forces applied to spindles [80]. We found that the rate and degree of spindle elongation as well as final spindle pole positions were nearly identical to wild type in both *par-2* and *par-6* heterozygotes (Fig 6A and 6B). The only visible difference was a reduction in the magnitude of posterior spindle pole oscillations in *par-2* heterozygotes (Fig 6C–6E).

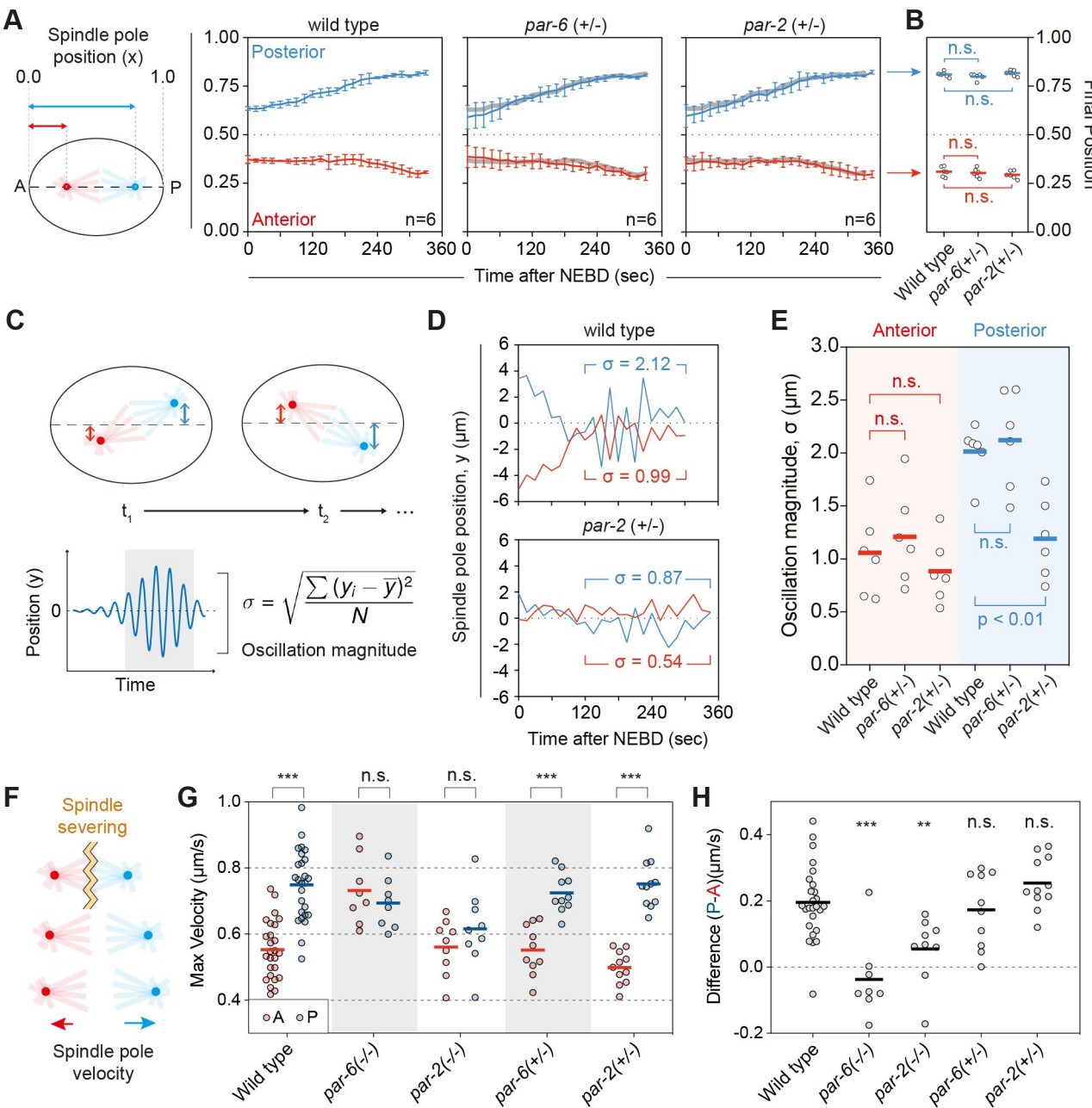

**Fig 6. Spindle positioning is highly robust to PAR dosage changes. (A)** Relative position of the anterior and posterior spindle poles along the A-P axis (x) from NEBD through telophase. Mean behavior for wild-type embryos shown as gray lines in *par-2(+/-)* and *par-6(+/-)* heterozygote plots. Note nearly identical behavior in wild-type and heterozygous embryos. Mean ± SD shown (*n* = 6 embryos, all conditions). **(B)** Comparison of final spindle pole positions taken at telophase from experiments in (A), defined as the time when the cleavage furrow was 50% ingressed and spindle poles exhibited no further outward motion. Individual data points and mean indicated. One-way ANOVA (vs. relevant wild type), Dunnett's correction. **(C)** Schematic for quantifying spindle oscillations. Oscillation magnitude (σ) was defined as the standard deviation of measured spindle pole displacement (y) off the central A-P axis (y = 0) from prometaphase to telophase. **(D)** Sample plots of spindle pole position for wild type and a *par-2(+/-)* embryos shown alongside oscillation magnitude (σ). **(E)** Heterozygous *par-2* embryos exhibit reduced spindle oscillations relative to wild-type and *par-6* heterozygotes; σ was similar for the anterior spindle pole across all 3 conditions. Individual data points and mean indicated. One-way ANOVA (vs. relevant wild type), Dunnett's correction. **(F)** Schematic of spindle severing experiments to detect changes in force applied to anterior and posterior spindle poles. **(G)** Max outward spindle pole velocity following spindle severing. As previously reported, *par-6(-/-)* leads to symmetric high and *par-2 (-/-)* leads to symmetric low pulling forces. Both heterozygotes show asymmetric pulling forces, similar to wild type. Mean and individual data points shown. Paired *t* test, Holm–Sidak correction. **(H)** Difference in max velocity (Posterior—Anterior) shown for samples in (G). Note that this difference is lost in the *par* null conditions but are not significantly different from wild type in either heterozygous condition. Mean and individual data points

shown. One-way ANOVA (vs. wild type), Dunnett's correction. The raw data underlying this figure can be found at https://doi.org/10.25418/crick.27153459.

As a more direct read-out of spindle pulling forces along the anterior-posterior (A-P) axis, we measured spindle pole velocities following ablation of the spindle midzone at anaphase onset (a.k.a. spindle severing; Fig 6F) [77]. Consistent with prior reports, PAR-2 and PAR-6 were both required for asymmetry in pulling forces, with loss of PAR-6 yielding symmetric high pulling forces, and loss of PAR-2 yielding symmetric low pulling forces. However, when we examined *par-2* and *par-6* heterozygotes, we found no detectable differences relative to wild type (Fig 6G and 6H), suggesting any changes in the magnitude of pulling forces in *par* heterozygotes are below the detection limits of this assay.

Taken together, our data suggest that the robustness of division size asymmetry to PAR dosage changes can be traced to robustness in the pattern and magnitude of forces driving spindle positioning along the A-P axis under such conditions. Thus, there must be substantial nonlinearity in the readout of local PAR concentrations by cortical force generators, effectively maintaining the asymmetry in pulling forces near wild-type levels in the face of approximately 50% reductions in a given PAR protein.

**Fate asymmetry.**   We next turned to the fate segregation pathway, which relies on the establishment of an anterior-to-posterior gradient of MEX-5/6 (Fig 7A). The MEX gradient is set up by a complementary gradient of PAR-1 kinase activity, which locally modulates the diffusivity of MEX proteins and is itself set up downstream of cortical polarity [71,81–83]. Once formed, the MEX gradient induces asymmetric segregation of fate determinants, including various germline markers such as PIE-1 and P granules as well as the cell cycle regulators that are responsible for division asynchrony [65,84–91].

Similar to our analysis of spindle positioning, we measured proximal features of the fate segregation pathway, including the PAR-1 and MEX-5 gradients. Whereas loss of PAR-6 or PAR-2 substantially reduced or eliminated PAR-1 asymmetry, asymmetry was normal in both *par-2* and *par-6* heterozygotes (Fig 7B). Consistent with the robustness of PAR-1 asymmetry in *par-2* and *par-6* heterozygotes, when we subjected embryos to progressive depletion of either protein, the MEX gradient was unchanged until dosage declined below 50% (Fig 7D and 7E). We conclude that the stability of PAR-1 asymmetry to changes in cortical PAR concentrations explains a substantial fraction of the robustness of fate asymmetry to intermediate changes in PAR dosage.

However, while this stability of PAR-1 asymmetry can explain the robustness of fate specification with respect to PAR-2 and PAR-6 dosage, it does not explain why fate is also relatively robust to depletion of PAR-1. Prior studies suggest that changes in PAR-1 concentration should directly impact the local kinase activity available to polarize MEX-5 [71]. We therefore scored both the PAR-1 and MEX-5 gradients as a function of PAR-1 dosage. As expected, the magnitude of the PAR-1 concentration difference across the zygote declined as a linear function of PAR-1 dosage (Fig 7C). By contrast, the MEX-5 gradient initially showed relatively modest changes in response to PAR-1 depletion, only decaying more strongly as depletion exceeded 50% (Fig 7F). Thus, similar to the stability of the PAR-1 gradient to changes in levels of its upstream regulators, PAR-2 and PKC-3, the MEX-5 gradient also appears robust to perturbation of its direct upstream regulator, PAR-1.

We therefore conclude that within the regime in which asymmetric division is robust to PAR dosage changes, both spindle and fate asymmetry pathways possess mechanisms to canalize phenotypic outputs in response to variation in local cortical PAR concentrations.

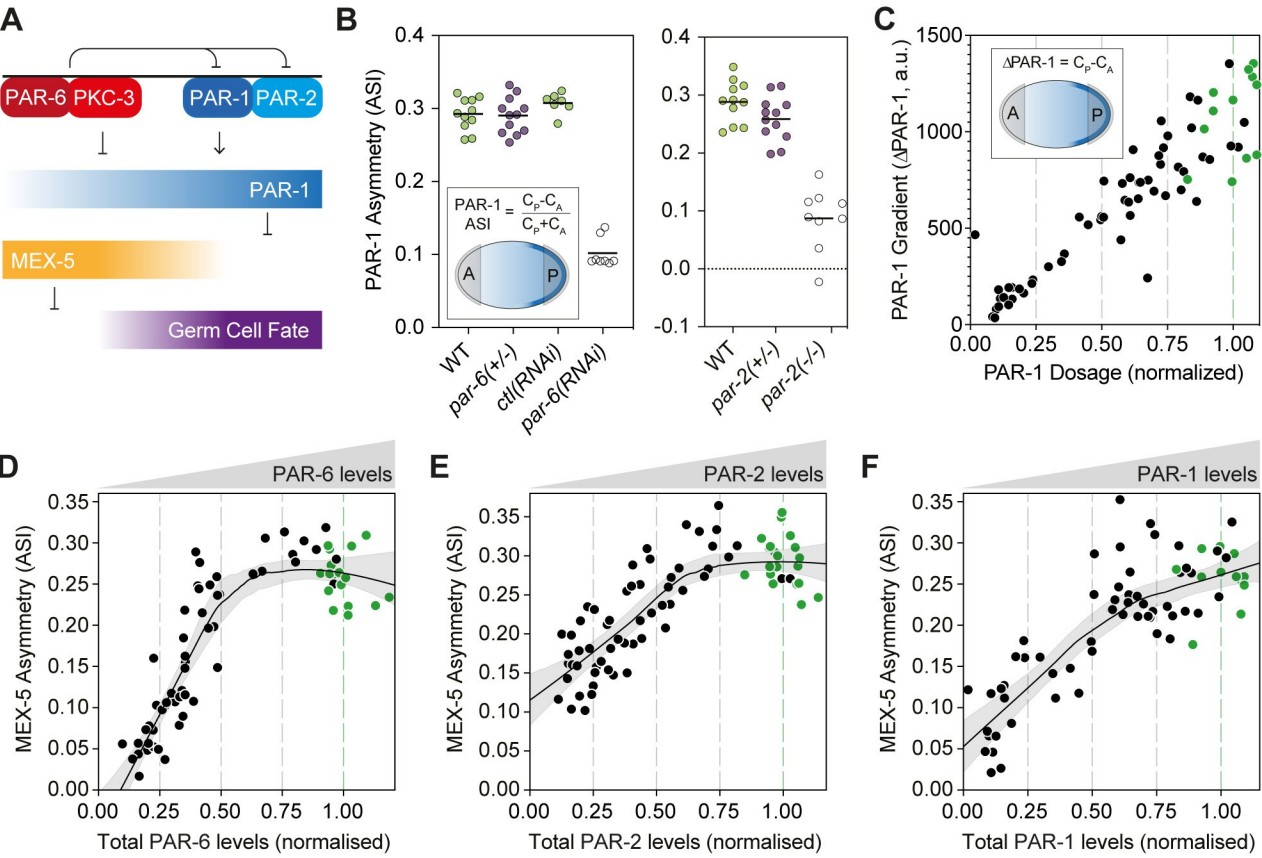

**Fig 7. Cytoplasmic asymmetry is robust to perturbations of PAR protein concentrations. (A)** Fate asymmetry is specified by a cytoplasmic gradient of MEX-5 that is downstream of PAR polarity. Cell cycle asynchrony is a commonly used proxy of fate asymmetry. Note that the mechanistic relationship and contributions of cortical vs. cytoplasmic PAR-1 asymmetry are not well understood. **(B)** PAR-1 gradient asymmetry (ASI) is robust in *par-2(+/-)* and *par-6(+/-)* heterozygotes. Genotypes indicated. *par-6(RNAi)* and *par-2(-/-)* homozygous mutants shown for comparison. **(C)** The absolute magnitude of the PAR-1 gradient (ΔPAR-1, arbitrary fluorescence units) declines near linearly with PAR-1 dosage. Magnitude of PAR-1 concentration difference between anterior and posterior ($C_P$-$C_A$) shown as a function of PAR-1 dosage. Note for (B, C), PAR-1 levels were the mean intensity in a region of interest containing both local membrane and cytoplasm regions—see Inset, Methods. **(D–F)** MEX-5 asymmetry responds nonlinearly to depletion of PAR proteins. MEX-5 asymmetry (ASI) as a function of dosage of PAR-6(D), PAR-2(E), and PAR-1(F). Fit lines in (D–F) indicate Lowess fit with 95% confidence interval determined by bootstrapping. In (C–F), wild-type data points are indicated in green. The raw data underlying this figure can be found at https://doi.org/10.25418/crick.27153459.

## PAR dosage alters sensitivity to symmetry-breaking cues

Our data so far suggest that robust division asymmetry emerges from 2 features: first, the stability of overall PAR asymmetry (but not concentrations) with respect to variations in PAR dosage, and second, readout mechanisms that insulate downstream pathway outputs from dosage-dependent variation in PAR concentrations at the membrane. We next turned our attention to the interpretation of upstream symmetry-breaking cues by the PAR network.

Polarization of PAR proteins in the zygote is triggered by several semi-redundant symmetry-breaking cues that impose asymmetry in PAR protein distributions. These asymmetries are then amplified and reinforced by self-organizing feedback to generate a stably polarized state (see Fig 1A). The dominant cue is cortical actomyosin flow, which is induced by the centrosome and advects aPAR proteins into the nascent anterior allowing pPAR proteins to invade the posterior membrane [53,92]. Coincidentally, a second, flow-independent centrosomal cue promotes pPAR loading onto the posterior pole that is amplified by concentration-dependent dimerization of PAR-2 [30,33,50]. Finally, several other cues have also been proposed that may

enhance the efficiency of symmetry-breaking, including curvature and hydrogen-peroxide produced by centrosome-associated mitochondria [26,93]. While the existence of multiple cues may help explain why symmetry-breaking is robust to loss of a given cue, the quantitative relationships between the strength of symmetry-breaking cues, PAR dosage/feedback, and the resulting pattern of PAR polarity have not been explored.

To determine the sensitivity of PAR polarity to cue strength under conditions of wild-type PAR dosage, we measured PAR-2 domain size as a function of cortical flow velocity, which was tuned through RNAi-mediated depletion of the myosin regulatory light chain, MLC-4. We found that domain size at NEBD was constant until cortical flow velocities were reduced by over 50% (below 0.05 µm/s). Beyond this point, domains became highly variable in size and position (Fig 8B). This increased variability likely reflects a transition to flow-independent symmetry-breaking pathways, which are associated with delays in domain formation, slow domain expansion, and failure to align polarity domains with the long axis [25,30,33,53,94]. Consistent with this interpretation, the observed threshold velocity of 0.05 µm/s roughly corresponds to the minimal cortical flow velocity required for symmetry-breaking when flow-independent cues are compromised (0.062 µm/s) [25]. Thus, while embryos require a minimal flow velocity to enter a flow-dependent polarization regime, once this is achieved, the degree of PAR polarity—here defined by PAR-2 domain size—exhibits minimal variability and thus is effectively decoupled from the strength of the symmetry-breaking cue (Fig 8B; flow velocities > 0.05 µm/s).

We next asked how the ability of the PAR system to respond to cues is affected by PAR dosage. Because polarization is generally robust to PAR dosage changes when wild-type cues are present, we used 2 suboptimal symmetry-breaking regimes: a flow-defective regime (*nmy-2 (ne3409)* or *mlc-4(RNAi)*) in which flows are absent and polarity is thought to rely on centrosomal microtubules [25,30,33] and a centrosome-defective regime (*spd-5(or213)* or *spd-5 (RNAi)*) in which the posterior centrosomal cue is compromised [26,56,95–100]. We found that control embryos with intact symmetry-breaking cues reliably formed PAR-2 domains despite reduction of PAR-2 dosage by up to approximately 75% relative to wild-type levels. By contrast, in both flow- and centrosome-defective conditions, PAR-2 domain formation failed when depletion exceeded 30% to 40%, marking a clear shift in the threshold level of PAR-2 required for efficient symmetry-breaking (Fig 8C–8F).

Corroborating our PAR-2 rundown experiments, PAR-2 domains were weaker and delayed in *par-2* heterozygous embryos subject to *mlc-4(RNAi)* compared to wild-type controls (Fig 8G and 8H). As we showed (see Fig 2C), *par-2* heterozygotes contain PAR-2 amounts that are roughly 60% to 70% of those found in wild-type embryos, which corresponds roughly to the point in the PAR-2 rundowns at which we began to observe polarization defects in cue-compromised embryos (Fig 8C).

Finally, to explicitly test the effects of both increased and decreased aPAR:pPAR ratios in cue-compromised conditions, we compared polarity outcomes in wild-type, heterozygous *par-2*, and heterozygous *par-6* embryos, depleted of the centrosome component SPD-5. Again, outcomes strongly depended on dosage (Fig 8I and 8J): in *spd-5(RNAi)* embryos with normal PAR dosage, all embryos exhibited clear PAR-2 domains, with a mix between monopolar embryos with a single PAR-2 domain and bipolar embryos with PAR-2 domains at both anterior and posterior poles (50% or less). Bipolar embryos are thought to arise from the loss of a single dominant centrosomal cue at one pole, which results in the embryo responding inappropriately to weaker cues at the anterior pole that are normally insufficient to induce PAR-2 domain formation [26,68]. In *spd-5(RNAi)* embryos heterozygous for *par-2*, a substantial fraction of embryos failed to exhibit clear PAR-2 domains, corroborating our RNAi rundown results that polarization in this regime is sensitive to PAR-2 dosage. At the same time, it was

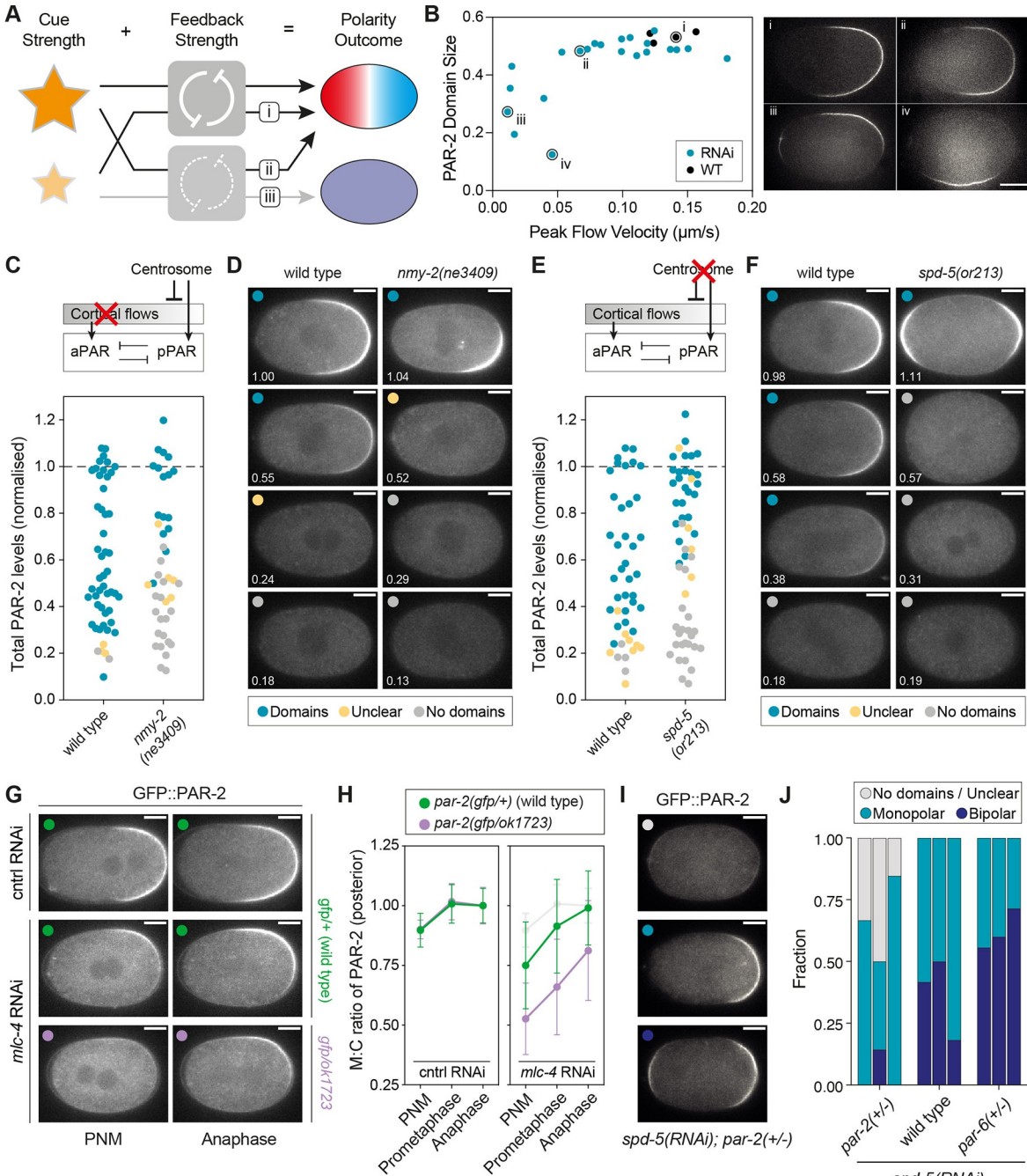

**Fig 8. PAR dosage reduction renders polarity sensitive to defects in polarity cues. (A)** Scheme for how reduced PAR dosage could sensitize embryos to compromised symmetry-breaking cues. At full strength, PAR feedback is sufficient to amplify signals provided by a weakened cue and thereby rescue normal polarity (i). Conversely, sufficiently strong cues can compensate for reduced PAR feedback to rescue polarity establishment in embryos partially depleted of PAR proteins (ii). However, in the presence of reduced PAR feedback, polarity becomes sensitive to cue strength (iii). **(B)** Above a threshold cortical flow velocity, PAR-2 domain size is nearly constant. Only upon progressive reduction in peak cortical flow velocity below a threshold value do PAR-2 domains undergo an abrupt shift to being variable sized and mispositioned, consistent with a shift to a flow-independent symmetry-breaking regime. Cortical flow velocities were reduced by *mlc-4(RNAi)*. Individual embryos are indicated and images of select examples shown at right. **(C, D)** Inhibition of cortical flow sensitizes symmetry-breaking to reduced PAR-2 dosage. GFP::PAR-2 dosage was progressively reduced by RNAi in wild-type and temperature-sensitive *nmy-2(ne3409ts)* embryos at the restrictive temperature (25°C) and embryos imaged just prior to NEBD. PAR-2 dosage was measured, embryos scored for the presence of GFP::PAR-2 domains, and the results plotted in (C). Example embryos at different PAR-2 dosages and exhibiting different phenotypes shown in (D). **(E, F)** Disruption of the centrosome cue sensitizes symmetry-breaking to PAR-2 dosage. Performed as in (C, D), but using the temperature-sensitive allele *spd-5(or213)*. Embryos were scored as

exhibiting domains if they exhibited clearly defined PAR-2 domains. Note that *spd-5(or213)* embryos often exhibit bipolar PAR-2 domains, which were scored as having domains for the purposes of this assay (see I, J). **(G, H)** Heterozygosity for *par-2(ok1723)* substantially delays symmetry-breaking in *mlc-4(RNAi)* embryos with reduced flows. Images (G) and quantification of membrane: cytoplasm ratio (H) at pronuclear meeting (PNM), prometaphase, and anaphase for *par-2(gfp/+)* or *par-2(gfp/ok1723)* embryos subject to either control or *mlc-4(RNAi)*. Control data shown in light gray in mlc-4(RNAi) panel for comparison. Note *par-2(gfp/+)* embryos were used as controls so that quantification would not be affected by differing levels of GFP signal. **(I, J)** Symmetry-breaking is dosage sensitive in centrosome-compromised embryos. *spd-5(RNAi)* embryos with the indicated *par* genotypes were scored for the presence or absence of clearly defined GFP::PAR-2 domains and whether 1 (monopolar) or 2 (bipolar) domains were present (J). Columns indicate frequency distributions for each genotype obtained in 3 replicate experiments (*par-2(+/-)*, *n* = 9, 14, 13; wild type, *n* = 12, 9, 11; *par-6(+/-)*, *n* = 9, 5, 7). Note the first replicate was performed at the semi-restrictive temperature of 20˚C, while the remaining 2 were performed at the fully restrictive temperature of 25˚C. All show a similar trend. Example images of scored phenotypes in (I). Scale bars, 10 μm. The raw data underlying this figure can be found at https://doi.org/10.25418/crick.27153459.

striking that among embryos with a PAR-2 domain, bipolarity was rare, consistent with reduced pPAR:aPAR ratios limiting the ability of the system to respond to multiple competing suboptimal cues. By contrast, *par-6* heterozygotes exhibited the opposite trend—all embryos exhibited clear PAR-2 domains, but the majority of embryos were bipolar. Thus, strong cues effectively mask an underlying sensitivity of the symmetry-breaking process to PAR dosage.

## Discussion

Here, we show that the robustness of asymmetric division in the *C. elegans* zygote can be traced to nonlinear signal-response relationships, which are a core feature of the modules that make up the asymmetric division pathway. In particular, we find that division asymmetry remains substantially normal in the face of approximately 50% reductions in the amount of any given component of the PAR protein network, which is the key regulator of both size and fate asymmetry.

### Robustness, heterozygosity, and nonlinear dose-response curves

Understanding the robustness of biological systems to approximately 2-fold changes in protein dosage has its roots in Mendel's observations of genetic dominance [101]. Genetic dominance describes the observation that the vast majority of wild-type alleles are "dominant" over loss-of-function alleles—i.e., most genes do not exhibit haploinsufficiency [102,103]. It has typically been attributed to the insensitivity of biochemical networks to 2-fold changes in enzyme dosage [104,105]. This insensitivity can emerge as a natural consequence of limiting substrate conditions or kinetic-linking of multi-enzyme reaction pathways, which give rise to nonlinear activity-flux relationships characterized by substantially flat regimes within which systems are effectively insensitive to enzyme dosage [13,105].

Implicit in such models is that such systems should also be particularly robust to overexpression [104]. Consistent with this picture, sensitivity to 2-fold reductions in dosage (i.e., haploinsufficiency) in *Saccharomyces cerevisiae* tends to result from the relatively rare cases in which both over- and under-expression impose substantial fitness costs—in other words, cases in which activity must be balanced [106]. The process of PAR-dependent asymmetric division therefore presents a paradox—it is sensitive to balance between anterior and posterior PAR proteins, yet simultaneously exhibits significant robustness to approximately 2-fold dosage changes.

What do we mean when we say that this system is robust? First, robustness only appears to extend to approximately 2-fold dosage changes—beyond this threshold, division asymmetry is progressively lost. Second, even within the "robust" regime, if one looks at the precision of the PAR pattern itself, such as local PAR concentrations, PAR domain size, or the number of PAR

domains, rather than overall asymmetry, the PAR network **is** sensitive to dosage changes (Figs 4 and 5). Yet, when we ask how polarity is interpreted, it is apparent that downstream processes are insensitive to substantial variation, at least within regimes spanning approximately 50% changes in protein dosage in which PAR proteins remain grossly asymmetric (Figs 3, 6, and 7). Thus, the networks governing asymmetric division must exhibit properties that ensure wild-type embryos occupy a regime in which the dosage-phenotype relationship is locally flat.

The consequence of this nonlinear signal processing is that subcritical perturbations will tend to be canalized, allowing embryos to achieve near wild-type levels of accuracy in the face of variation in the function or activity of any given module. This "decoupling" between signals and outputs [5] means that measures of division asymmetry, such as the size and fate of daughter cells, are not strongly affected by the strength of symmetry-breaking cues or specific features of the PAR pattern such as domain size, PAR boundary position, or local PAR protein concentrations. Thus, in contrast to models of position specification by gradients, the PAR pattern does not appear to encode concentration-dependent positional information [107,108]. Rather, there is a more steplike collapse that occurs only once perturbations exceed a threshold. We hypothesize that this design feature effectively insulates a given "module" from the effects of genetic, environmental, or stochastic variation arising in other modules. Propagation of variance is thereby minimized to maintain embryos along the correct developmental trajectory, leading to stability of phenotypic outcomes.

## Encoding robustness in symmetry-breaking

A key question that remains is how such nonlinear input:output mapping is encoded. One explanation is the feedback-driven self-organizing nature of the PAR network itself. We previously proposed a model for symmetry-breaking in this system that involves a cue-driven switch between stable unpolarized/uniform and polarized/patterned states defined by bistable reaction kinetics [25,53,54]. This relatively simple modeling framework certainly underestimates the extent of self-organization present in vivo, which includes both positive and negative feedback within the PAR network as well as the boundary stabilizing effects of PAR-dependent patterning of actomyosin flows, making precise quantitative comparisons difficult [48–50,52,69]. Nonetheless, key qualitative features of the model help explain the robustness of symmetry-breaking we observe.

First, the notion of polarity as a triggered process of self-organization implies that the final polarized state will be defined primarily by intrinsic, parameter-dependent features of the patterning network. In this paradigm, symmetry-breaking cues need only to impart the system with sufficient asymmetry to push the initially unpolarized system across a transition point beyond which PAR network feedback takes over and drives the system towards the polarized state [25]. Consequently, the polarized state is expected to be robust to the nature of the cue, provided it is sufficient to "flip" the system, similar to the relationship we observe between PAR-2 domain size and cortical flow velocity (Fig 8B).

Second, a generic feature of this class of models is that the stability of uniform states to perturbation with respect to PAR dosage takes the form of a cusp bifurcation, with changes in PAR dosage effectively tuning the responsiveness of the system by moving closer or further away from the bifurcation [53,54]. Given an initial aPAR-dominant, unpolarized state, depletion of PAR-2 tends to move the system away from the bifurcation, reducing responsiveness to cues, while depletion of PAR-6 does the reverse. Normal symmetry-breaking cues appear sufficiently strong to mask this potential effect of PAR dosage on symmetry-breaking in otherwise wild-type embryos—cues are simply stronger than they need to be (Fig 8) [25]. However, when the strength of cues is reduced, the expected dependency on PAR dosage is revealed:

reduction in PAR-2 dosage reduces the probability of polarization, while reduction in PAR-6 enhances it, but with the consequence that the system responds inappropriately to multiple cues leading to bipolarity (Fig 8I and 8J). Such behavior may account for the reported sensitivity of polarity phenotypes to expression of ectopic transgenes or hypomorphic alleles in cue-compromised embryos [26,30,33,56,68,96,100,109].

## Robust readout of PAR polarity

While a paradigm of triggered self-organization helps explain the observed robustness of PAR patterning to variation in symmetry-breaking cues, the design features that provide for robustness in the interpretation of variable PAR concentration profiles by downstream pathways are less clear. It is striking how little these processes are affected by real and substantial alterations in PAR concentrations given that spindle pulling forces and cytoplasmic gradients of fate determinants are thought to be directly regulated by the PAR kinases PKC-3 and PAR-1. As the precise mechanisms by which PAR signals are transduced to downstream spatial pathways remain enigmatic, a full exploration of the features in these pathways that give rise to nonlinear dosage-phenotype relationships is beyond the scope of this work. Nonetheless, our data imply nontrivial complexity in the readout of PAR concentrations, either in the mapping of concentration to kinase activity or in the interpretation of gradients of PAR activity by downstream pathways.

A disconnect between activity and concentration could help explain resilience to dosage changes. Indeed, in the case of PAR-1, there is reason to believe the concentration-activity relationship may not be straightforward. First, there are distinct membrane and cytoplasmic pools of PAR-1, the relative role of which is unclear [71,110]. Second, the gradient of MEX-5 appears sharper than expected given the rather shallow cytoplasmic PAR-1 concentration gradient [71]. Finally, there is data suggesting that a PAR-1 activity gradient can persist even in the absence of a visible PAR-1 concentration gradient [42].

Metabolic control theory provides another potential explanation. As we note above, specific features of an enzymatic network, such limiting substrate conditions or kinetic linking of multi-enzyme reaction pathways, can provide for regimes that are insensitive to enzyme activity [13,105]. However, it is unclear whether we can draw parallels between such models and the regulation of division asymmetry by the PAR network. Neither PAR-1 nor PKC-3 are present at substantially greater concentrations than their respective substrates and it is unclear if we can equate the dosage insensitivity of metabolic flux in metabolic networks to the spatio-temporal regulation of molecules by kinase-phosphatase circuits, particularly one that is, at least at some level, sensitive to balance.

One obvious limitation in delineating the molecular mechanisms underlying the robustness we observe comes from our lack of knowledge regarding system-level integration of the various potential signals provided by the PAR proteins and the responses of effector pathways. This is most obvious for spindle positioning where multiple kinases (PKC-3, PAR-1) impact the localization and/or activity of various regulators of spindle pulling forces, including key components of cortical dynein complexes LIN-5, GPR-1/2, as well as at least 1 spatial regulator, LET-99, that is thought to suppress lateral pulling forces [70,73,74,111]. Spindle position further appears less sensitive to PAR-1 than PAR-2 (Fig 3H), counterintuitively suggesting a more direct role for the scaffold (PAR-2) than the kinase (PAR-1) in regulating spindle forces, an interpretation that would be consistent with quantitative trait mapping that linked *par-2* with pulling force/spindle size [112]. We would also note that our data is also consistent with genetic data showing that *par-1* mutants often show residual size asymmetry [37,72]. Finally, we cannot ignore features of the spindle positioning network, itself. Spindle oscillations are

curious given their lack of obvious functional role and their sensitivity to dosage of PAR-2 and force generator components (Fig 6E) and [80]. While it is often proposed that oscillations may constitute an epiphenomenon related to the underlying mechanisms of spindle positioning [40,80], the contrast between sensitivity of oscillations and the stability of spindle positioning [80,113,114] makes it tempting to speculate that they may be related to buffering excess pulling force, thereby stabilizing pulling forces in the A-P direction. That said, excess forces do not universally enhance oscillations and appear uniquely present with the *Caenorhabditis* clade despite the widespread prevalence of asymmetric spindle positioning in nematodes. Thus, the precise relationship between force, oscillations, and precision of spindle positioning remains unclear [20,23,40,114], particularly given that additional positional control mechanisms are likely at play that stabilize final spindle position even under regimes in which pulling forces acting along the A-P axis are visibly altered—a regime beyond what is considered here [80,112,115,116].

Ultimately, resolving the molecular origins of robustness will require novel approaches to resolve the quantitative features of the signal-response dynamics reported here. For example, we currently lack appropriate tools to map the in vivo concentration-activity relationship for the key kinases in the system, including direct readouts of kinase activity or substrate phosphorylation. We also generally lack quantitative characterization of core molecular circuits, for example, in the response of substrates to kinase activity, though increased use of quantitative perturbation and modeling as well as progress in reconstituting key processes in vitro are enabling progress on this front [25,48–50,52,55,69,117–123].

In conclusion, through establishing quantitative perturbation-phenotype maps for the first embryonic cell division of *C. elegans*, we have revealed how nonlinear responses to spatial information drive robust developmental outcomes within an asymmetric division program. How these nonlinearities are encoded remains a key area for future investigation.

## Methods

### *C. elegans* strains and maintenance

*C. elegans* strains were maintained on OP50 bacterial lawns seeded on nematode growth media (NGM) at 16˚C or 20˚C under standard laboratory conditions [124]. Worm strains were obtained from *Caenorhabditis* Genetics Center (CGC) and are listed in S1 Table. Note that analysis of zygotes precludes determination of animal sex.

### *C. elegans* husbandry and generation of heterozygous genotypes

In some cases, heterozygotes were selected from stable lines for analysis (Figs 3A, 3B, and 6A–6E). To generate fluorescently tagged heterozygous animals harboring one mutant allele copy, males of relevant balanced strains were crossed into hermaphrodite L4 larvae of homozygous FP-tagged strains (see Figs 2B–2D, 3C, 3D, 3F, 3G, 7B, and 8G–8J). Similarly, to obtain heterozygous worms where the 2 alleles of a given gene are tagged with different FPs, males of a given homozygous background (e.g., mCherry-tagged gene) were crossed into hermaphrodite L4 larvae of a homozygous strain expressing the relevant gene labeled with a different FP (e.g., GFP) (see Figs 2D and S1A–S1E). To measure the dependence of PAR-1 polarity on PAR-2 and PAR-6 levels, animals of heterozygous or null genotypes were directly selected from balanced strains expressing PAR-1::GFP prior to sample preparation and imaging (see Fig 7B). For spindle severing experiments (Fig 6F–6H), heterozygous *par-2(+/-)* and homozygous *par-2(-/-)* animals were obtained from NWG0632, with XA3501 serving as the wild-type control. Heterozygous *par-6(+/-)* and wild-type *par-6(+/+)* animals were obtained sorting F1 progeny of a XA3501 x NWG0141 cross, with *par-6(RNAi)* animals serving as the *par-6(-/-)* condition.

## RNAi culture conditions

RNAi by feeding was performed according to previously described methods [125]. Briefly, HT115(DE3) bacterial feeding clones were inoculated from LB agar plates to LB liquid cultures and grown overnight at 37˚C in the presence of 50 μg/ml ampicillin (until a turbid culture was obtained). To induce high dsRNA expression, bacterial cultures were then treated with 1 mM IPTG before spotting 150 μl of culture onto 60 mm NGM agar plates (supplemented with 10 μg/ml carbenicillin, 1 mM IPTG) and incubated for 24 h at 20˚C. In general, to obtain strong/complete gene depletion (or allele-specific depletion in the case of experiments using dual-labeled alleles), L3/L4 larvae were added to RNAi feeding plates and incubated for 24 to 36 h depending on gene and temperature (see Figs 2D, 2G–2I, 7B, 8G–8J, S1 and S2A–S2E). To perform protein rundowns, where dosage is depleted from wild-type concentration through full depletion, L3/L4 larvae were either: placed on relevant RNAi for variable periods of time (from 4 h to 24–36 h incubation) or placed on plates where the relevant RNAi feeding clone was mixed with a control RNAi clone (expressing non-targeting dsRNA) to dampen the strength of protein depletion (see Figs 2E, 2F, 3C–3H, 4, 5, 7C–7F, 8B, S2A, S2B, and S2F–S2H). For the protein rundowns using temperature-sensitive strains, L3/L4 larvae were placed on RNAi for variable amounts of time (or on plates containing par RNAi mixed with control RNAi, as detailed above) at 16˚C, and then placed on OP50 and shifted to 25˚C for 75 min (see Fig 8C and 8D) or 45 min (see Fig 8E and 8F) prior to imaging.

## *C. elegans* dissection and mounting for microscopy

For most experiments (namely, for Figs 2B–2D, 3A–3D, 3F–3G, 6A–6E, 7D–7F, 8B, S1, and S2A–S2E), early embryos were dissected from gravid hermaphrodites in 5 to 6 μl of M9 buffer (22 mM KH$_2$PO$_4$, 42 mM NaHPO$_4$, 86 mM NaCl, and 1 mM MgSO$_4$) on a coverslip and mounted under 2% M9 agarose pads [126]. In some instances (see Figs 2E–2J, 3E, 3H, 4, 5, 6G, 6H, 7B, 7C, 8C–8J, and S2F–S2H), to minimize eggshell autofluorescence that can be prominent with agarose mounts, embryos were dissected in 8 to 10 μl of egg buffer (118 mM NaCl, 48 mM KCl, 2 mM CaCl$_2$, 2 mM MgCl2, 25 mM HEPES, pH 7.3) and mounted with 20 μm polystyrene beads (Polysciences, Inc.) between a slide and coverslip as in [127].

## Microscopy

Unless specified otherwise, midsection confocal images were captured on a Nikon TiE with a 60×/1.40 NA oil objective, further equipped with a custom X-Light V1 spinning disk system (CrestOptics, Rome, Italy) with 50 μm slits, Obis 488/561 fiber-coupled diode lasers (Coherent, Santa Clara, California, United States of America) and an Evolve Delta EMCCD camera (Photometrics, Tucson, Arizona, USA). Imaging systems were run using Metamorph (Molecular Devices, San Jose, California, USA) and configured by Cairn Research (Kent, United Kingdom). Filter sets were from Chroma (Bellows Falls, Vermont, USA): ZT488/561rpc, ZET405/488/561/640X, ET535/50m, ET630/75m.

To obtain confocal images of samples expressing mNG-tagged genes, imaging was performed on a Leica TCS SP8 inverted microscope (Leica Microsystems Ltd, Wetzlar, Germany), equipped with an Apo CS2 63x/1.40 NA oil objective and a HyD detection system. Imaging was managed with LAS X software (Leica Microsystems Ltd, Wetzlar, Germany), and acquisition was set at a scanning speed of 400 Hz with pinhole aperture set to 2 AU. Unlike the Nikon configuration detailed above, this microscope offered the capability of imaging samples with either 488-nm or 514-nm excitation, thus permitting the distinction between GFP and mNG specific fluorescence (as in the case of experiments shown in Figs 2D, S1D–S1F, and S2C–S2E).

For experiments shown in Fig 8B, embryos were imaged on an Olympus IX71 equipped with a Yokogawa spinning disk head using a 63x/1.40 Oil UPlanSApo objective, 488, 561-nm DPSS lasers, an iXon EMCCD camera and ImageIQ (Andor Technology). Images were initially captured in the GFP channel at 2 s intervals covering the polarity establishment phase to measure cortical flow rates. A single snapshot in the mCherry channel was then captured at NEBD to quantify PAR-2 domain size.

For most experiments (see Figs 2, 4, 5, 7B, 7C, 8C–8F, 8I, 8J, S1, and S2), image acquisition was performed by taking still images of live embryos at approximately NEBD. Data were acquired with both fluorescence and transmitted light configurations. In all experiments, imaging was done at 20˚C, except in the case of temperature-sensitive alleles, where acquisition was done at 25˚C using an objective temperature control collar (Bioptechs) (see Fig 8C–8F). In some instances (see Fig 3C–3H), the imaging pipeline was expanded to include analysis of the embryonic 4-cell stage. For this, time series of embryo development were acquired (at 15 s interval) as the AB and P1 blastomeres underwent cell division.

To image zygotic polarization in the various dosage/cortical flow regimes shown in Fig 8G and 8H, embryos were imaged from pronuclear migration/early prophase through telophase/cytokinesis at 60 s intervals.

For data shown in Fig 3A and 3B, where unlabeled lines were used, embryos were imaged using transmitted light only (i.e., DIC). In this case, data acquisition was only done at the 2-cell (snapshot taken at "birth" of AB and P1 cells) and 4-cell stage development (performing time series as detailed above).

To image spindle dynamics in mitotic zygotes (as detailed in Fig 6A–6E), samples were filmed from NEBD through telophase at 15 s intervals using DIC.

Spindle severing (Fig 6G and 6H) was performed at anaphase onset as judged by the initial separation of chromosomes using a 355 nm pulsed UV laser directed via an iLAS Pulse unit (Roper). An approximately 15 μm line traversing the spindle midzone was typically used as the ablation ROI. GFP channel images were captured using a 100 ms exposure stream acquisition and ablation triggered using the LiveReplay function in Metamorph.

## Image processing prior to fluorescence quantitation

For most experiments, and in order to quantify fluorescence signal and gauge protein dosage (total and/or cortical concentrations), images of embryos were taken alongside a local background image (with no samples in the field of view), which was subtracted from the image prior to analysis. Note that this step can usually be omitted without much detriment; however, background subtraction may improve images in cases where the background signal is uneven or variable.

## Image analysis—Quantitation of total and cortical fluorescence

For quantification of whole-embryo fluorescence intensities (to account for protein levels in both cytoplasm and membrane/cortex), mean pixel intensity was measured within a manually defined region of interest (ROI) encompassing the entire cross-section of the embryo.

To measure cortical concentrations, a 50-pixel-wide (12.8 μm) line following the membrane around the embryo was computationally straightened, and a 20-pixel-wide (5.1 μm) rolling average filter was applied to the straightened image. Intensity profiles perpendicular to the membrane at each position were fit to the sum of a Gaussian component, representing membrane signal, and an error function component, representing cytoplasmic signal. Membrane concentrations at each position were calculated as the amplitude of the Gaussian component. Cortical levels in Figs 4, 5, and S2 were calculated as the mean membrane concentration in the

posterior-most (PAR-1, PAR-2), or anterior most (PAR-3, PAR-6, PKC-3) 33% of the embryo perimeter. This protocol is similar to previously published methods [25,68] with relevant code available at https://doi.org/10.25418/crick.27153459.

For quantification of fluorescence of GFP- or mNG-tagged proteins (when using 488-nm excitation and a 535/50 nm emission filter), we applied SAIBR for autofluorescence correction [60]. This protocol was established to circumvent the high autofluorescence emission that results from 488-nm excitation (the most commonly used wavelength when imaging green fluorophores). This method exploits the fact that autofluorescence typically has a much wider emission spectrum than GFP. The protocol involves the use of a parallel channel, with a red-shifted emission filter (namely, a 630/75 nm filter), that is used to gauge the level of autofluorescence in the sample. The inferred autofluorescence is then subtracted from the measured signal intensity in the GFP channel (which is, in essence, a sum of both fluorophore-specific and nonspecific signals), thus yielding a more accurate estimate of protein concentrations. Subtraction can be done on a pixel-by-pixel basis (allowing for spatial signal correction) or on an embryo-by-embryo basis (e.g., to quantify whole-embryo fluorescence intensities, as indicated above). Note that SAIBR is also compatible when quantifying GFP fluorescence in embryos that co-express additional, spectrally distinct fluorophores (such as mCherry), as detailed in [60].

To quantify fluorescence in embryos expressing mNG-tagged PARs when using a 514-nm excitation and 550/50 nm emission filter configuration which minimizes autofluorescence, hereafter referred to as "mNG channel," a mean background signal was first measured across a sample of unlabeled embryos, and this mean value was then simply subtracted from the mNG channel signal in the mNG-tagged embryos. A similar protocol was employed in the quantitation of mCherry-tagged PAR signal, but in this case utilizing a channel with 561-nm excitation and a 630/75 nm emission filter.

### Image analysis—Dependence of PAR-2 polarity on cortical flows

For this experiment, embryos expressing NMY-2::GFP and mCherry::PAR-2 (TH306) were imaged on an Olympus IX71 spinning disk microscope (details above). Cortical flow velocities were extracted from cortical GFP time series data (2 s interval) by particle image velocimetry using the Matlab *mpiv* package (http://www.oceanwave.jp/softwares/mpiv) using custom scripts. PAR-2 domain size was quantified from midplane images taken at NEBD (see Fig 8B).

### Image analysis—Scoring polarity establishment under cue/dosage perturbations

To assess polarity establishment and PAR-2 domain formation (Fig 8C–8F, 8I and 8J), zygotes were scored as follows: "Domains," where a PAR-2 domain is visibly formed at NEBD (or early mitosis); "Unclear," where no membrane domain is achieved, but where there appears to be marginal PAR-2 cortical enrichment; and "No domains," for embryos that have clearly failed to break symmetry. In the case of *spd-5* embryos, domain-containing embryos were subdivided into "Monopolar" and "Bipolar." Images were scored independently by 2 researchers with disagreements settled through tie-break scoring by a third researcher.

### Image analysis—Measurement of membrane:cytoplasm signal ratio

To gauge cortical loading of PAR-2 in embryos at different stages of mitosis (see Fig 8H), analysis was done as follows: an intensity profile was obtained by drawing a 10-pixel-wide (2.6 μm) line perpendicular to PAR-2 domain (bisecting the center of domain) from inside to outside of embryo. With this linescan, a membrane signal was calculated by averaging the peak 5 pixels

of the profile (to cover the approximate membrane thickness). In parallel, to obtain a cytoplasmic signal, mean pixel intensity was measured within a 900-pixel rectangular ROI drawn in the embryo interior (excluding nuclear/spindle area). These 2 values were then used to calculate a membrane:cytoplasm ratio. This analysis was done using simple, custom Fiji (https://imagej.net/software/fiji/, [128]) and Matlab scripts to aid automation.

### Image analysis—Measurement of 2-cell size asymmetry

Area of AB and P1 blastomeres was measured on midplane cross-sections of 2-cell embryos, using manually defined ROIs or through semi-automated segmentation. The size asymmetry was then calculated by dividing the area of AB by the area of the whole embryo. Analysis was performed with custom macros in Fiji (see Figs 3A, 3F–3H, S2G, and S2H).

### Image analysis—Spindle dynamics

Time series of mitotic zygotes were acquired using DIC microscopy, where samples were imaged from NEBD through telophase at 15 s intervals. Centrosome coordinates were tracked manually using semi-automated code in Fiji. Spindle pole position was determined by measuring the distance of the anterior and posterior centrosomes to the anterior pole of the embryo (Fig 6A). Final centrosome position was defined at telophase as shown in Fig 6B. To monitor transverse oscillations (as seen in Fig 6C–6E), spindle pole position (y) was measured relatively to the nearest point on the central A-P axis (defined as y = 0). Oscillation magnitude was defined as the standard deviation of the set of centrosome positions (relative to A-P axis) measured from late prometaphase through telophase.

For spindle severing experiments, spindle pole position was tracked in Fiji using Trackmate [129] and the output positions exported for further analysis in Matlab. Tracks of x and y position were smoothed over 10 frames (1 s). Displacements were measured for rolling 1 s intervals and the maximum value used to calculate the maximum velocity. For purposes of analysis, wild-type embryos from XA3501 (+/+), XA3501 (+/+, control RNAi), and F1 (+/balancer) animals from an XA3501 x NWG0141 cross were pooled.

### Image analysis—Cytokinesis asynchrony in AB and P1 blastomeres

The time lag between AB and P1 divisions was measured in DIC time series of live embryos (Fig 3C–3E). More specifically, asynchrony was defined as the period between the furrow ingression of both blastomeres ($t_{AB}$ and $t_{P1}$—see illustration in Fig 3B). Furrowing was timed at the point of cortical indentation immediately preceding membrane "folding."

### Image analysis—PAR asymmetry and cytoplasmic ASI

For cortical PAR asymmetry, we define a signal normalized PAR asymmetry according to the following formula, which takes into account combined aPAR and pPAR signal:

$$\frac{(A_A - A_P) - (P_A - P_P)}{A_A + A_P + P_A + P_P}$$

Here, $A_A$ and $P_A$ are the concentrations of aPAR proteins in the anterior and posterior, respectively, and $A_P$ and $P_P$ are the corresponding concentrations of pPAR proteins. Note all concentrations are normalized to the peak concentrations achieved in wild-type embryos. Effectively, this measure defines integrated PAR asymmetry as the sum of the differences in aPAR and pPAR proteins at the 2 poles, normalized to total signal.

Asymmetry of PAR-1 and MEX-5 patterning was measured in midplane images of zygotes expressing the relevant FP-tagged gene at NEBD. Semi-circle ROIs were drawn on opposite sides of the embryo, and mean pixel intensity for anterior (A) or posterior (P) were retrieved accordingly. ASI was then defined as $\frac{A-P}{A+P}$ or $\frac{P-A}{A+P}$ for MEX-5 and PAR-1, respectively, such that ASI is always positive. For quantifying the response of the PAR-1 gradient to PAR-1 depletion, the gradient magnitude (ΔPAR-1) is reported as the absolute concentration difference (P-A) to capture the change in magnitude of the gradient upon PAR-1 dosage reduction. As the relative contributions of membrane and cytoplasmic populations of PAR-1 towards MEX-5 asymmetry are unclear, the semi-circular ROIs were chosen to include both membrane and cytoplasm signals.

## Statistics and formal analysis

Dosage versus phenotypic variance plots (Fig 3C–3H, "Var") were generated using a moving-window method. For each data set, a Gaussian weighting function (half-width = 0.1) was moved along the dosage axis, and Gaussian-weighted mean dosage was plotted against Gaussian-weighted phenotypic variance; 95% confidence intervals were determined by bootstrapping.

Unless otherwise specified, statistical analysis was performed in Prism (Graphpad).

## Supporting information

**S1 Fig. Lack of homeostatic dosage compensation of *par* gene expression in heterozygous animals. (A)** Schematic for dosage compensation assay using allele-specific RNAi-depletion. **(B, C)** PAR-6::GFP (B) and GFP::PAR-2 (C) levels for *gfp/gfp*, *gfp/mCherry*, and *gfp/mCherry (RNAi)* genotypes (left) together with controls for the depletion of mCherry-tagged alleles by RNAi (right). Note that compensation, if present, should be manifest as a difference in GFP levels in *gfp/mCherry* embryos ± *mCherry(RNAi)*. *gfp/gfp* and *+/+* embryos are shown to control for bleedthrough into the mCherry channel and to confirm zero point, respectively. **(D–F)** mNG::PKC-3 (D), mNG::PAR-3 (E), PAR-1::mNG (F) levels for the indicated genotypes (left) together with controls for depletion of GFP-tagged alleles by RNAi (right). Note that compensation, if present, should be manifest as a difference in mNG levels in *mNG/gfp* embryos ± *gfp (RNAi)*. Statistics, unpaired *t* test. *gfp/gfp* and *+/+* embryos are shown as controls for specificity of mNG excitation and to confirm zero point, respectively. The raw data underlying this figure can be found at https://doi.org/10.25418/crick.27153459.
(TIF)

**S2 Fig. Overall polarity is robust despite sensitivity of the PAR network to dosage changes. (A, B)** Cortical concentrations of PAR-6::GFP(A, NWG0119) and GFP::PAR-2(B, NWG0167) decline as a function of total dosage. **(C–E)** Cortical concentrations of PKC-3(C), PAR-3(D), and PAR-1(E) are reduced approximately 50% in heterozygous (*mNG/-*) embryos relative to homozygous (*mNG/mNG*) controls. Colored data points in (A–E) indicate homozygous (*xfp/xfp*, green) and heterozygous (*xfp/-*, purple) embryos. Note heterozygous conditions in (A–E) were *par-X(xfp₁/xfp₂(RNAi))* as in Fig 2D. **(F)** PAR asymmetry (ASI) as a function of total PAR-6 dosage as in Fig 4E, but highlighting individual embryos shown in (G). **(G)** Example embryos showing PAR-2 localization at NEBD, anaphase and cytokinesis for differing PAR-6 dosage. Dosage, PAR asymmetry, and size asymmetry are shown below each embryo. Embryos shown in (G) are depicted as red data points in (F, H). **(H)** Two-cell size asymmetry as a function of PAR asymmetry. Scale bars, 10 μm. The raw data underlying this figure can be found at

https://doi.org/10.25418/crick.27153459.
(TIF)

**S1 Table. Strains and reagents.**
(XLSX)

## Acknowledgments

We thank Buzz Baum, Michalis Barkoulas, Jacques Pecreaux, and the Goehring Lab for comments on the manuscript. Strains and/or reagents were graciously provided by Dan Dickinson, Ken Kemphues, and Geraldine Seydoux. Additional strains were provided by the Caenorhabditis Genome Center (CGC), which is funded by NIH Office of Research Infrastructure Programs (P40 OD010440), and the Mitani Lab via the National Bio-Resource Project of the MEXT, Japan. We also thank Tony Hyman and Stephan Grill at the Max Planck Institute of Molecular Cell Biology and Genetics, in whose laboratories some initial experiments were carried out.

## Author Contributions

**Conceptualization:** Nelio T. L. Rodrigues, Nathan W. Goehring.

**Formal analysis:** Nelio T. L. Rodrigues, Tom Bland, KangBo Ng, Nathan W. Goehring.

**Funding acquisition:** Nathan W. Goehring.

**Investigation:** Nelio T. L. Rodrigues, Tom Bland, KangBo Ng, Nathan W. Goehring.

**Methodology:** Nelio T. L. Rodrigues, Tom Bland.

**Project administration:** Nathan W. Goehring.

**Resources:** Nelio T. L. Rodrigues, Nisha Hirani.

**Writing – original draft:** Nelio T. L. Rodrigues, Nathan W. Goehring.

**Writing – review & editing:** Nelio T. L. Rodrigues, Tom Bland, KangBo Ng, Nisha Hirani, Nathan W. Goehring.

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
