## [Editor Report · Decision Letter 0]

15 Nov 2023

Dear Dr Goehring, 

Thank you for submitting your manuscript entitled "Nonlinear readout of spatial cues underlies robustness of asymmetric cell division" for consideration as a Research Article by PLOS Biology.

Your manuscript has now been evaluated by the PLOS Biology editorial staff as well as by an academic editor with relevant expertise and I am writing to let you know that we would like to send your submission out for external peer review.

Once your full submission is complete, your paper will undergo a series of checks in preparation for peer review. After your manuscript has passed the checks it will be sent out for review. To provide the metadata for your submission, please Login to Editorial Manager (https://www.editorialmanager.com/pbiology) within two working days, i.e. by Nov 17 2023 11:59PM.

Kind regards,

Ines

--

Ines Alvarez-Garcia, PhD

Senior Editor

PLOS Biology

---

## [Decision Letter · Decision Letter 1]

5 Feb 2024

Dear Dr Goehring,

Thank you for your patience while your manuscript entitled "Nonlinear readout of spatial cues underlies robustness of asymmetric cell division" went through peer-review at PLOS Biology and please accept my apologies again for the delay in providing you with our decision. Your manuscript has been evaluated by the PLOS Biology editors, an Academic Editor with relevant expertise, and by two independent reviewers.

The reviews are attached below. As you will see, the reviewers find the conclusions interesting, but they also raise several concerns that should be addressed to consider the manuscript for publication. Reviewer 1 asks for a couple of clarifications regarding the phenotype observed when PAR-2 is depleted, and also misses an analysis of the anterior and posterior pulling forces in the experiments on spindle positioning that could inform on the robustness of the spindle positioning pathway. This reviewer also would like further discussion taking into consideration the field of modelling PAR polarity in the C. elegans embryo, and questions to what extend the current models predict that changes in PAR protein level will lead to the observed effects of reducing cortical levels while maintaining overall polarisation. Reviewer 2 would also like you to clarify several points about the data and the way some of the experiments were performed, some of the terms used, and the meaning of several statements included in the results, among other issues.

In light of the reviews, we are pleased to offer you the opportunity to address the comments from the reviewers in a revision that we anticipate should not take you very long. We will then assess your revised manuscript and your response to the reviewers' comments with our Academic Editor aiming to avoid further rounds of peer-review, although might need to consult with the reviewers, depending on the nature of the revisions.

**IMPORTANT - SUBMITTING YOUR REVISION**

3. Resubmission Checklist

a) *PLOS Data Policy*

b) *Published Peer Review*

Sincerely,

Ines

--

Ines Alvarez-Garcia, PhD

Senior Editor

PLOS Biology

Reviewers' comments

Rev. 1:

In this manuscript the authors investigate to what extent PAR-dependent polarity establishment (PAR polarity itself and downstream size asymmetry and spindle positioning) in the one cell embryo of C. elegans is robust to changes in protein dosage, and what the sources of robustness are. The work is based on prior observations in the field that polarization is resilient to changes in temperature, pressure on the embryo, gene dosage, and symmetry breaking cues.

The manuscript begins by examining whether robustness to changes in par gene dosage are due to compensatory dosage regulation. The experiments are performed thoroughly and investigate 5 different par genes, making use of a recently optimized fluorescence quantification protocol also developed in the Goehring lab. While some dosage compensation is observed, protein levels in heterozygous animals as severely reduced, indicating that dosage compensation is not a major driver of robustness. Downstream readouts of PAR polarity, specifically division size asymmetry and spindle positioning, are also robust to changes in PAR protein dosage.

To investigate the source of robustness, the authors quantitatively measure cortical PAR protein levels and distribution upon reduction of overall PAR protein levels. Overall, while cortical PAR concentrations are affected, polarized PAR domains are observed until PAR protein levels are depleted below a threshold value, indicating that PAR polarity establishment itself is robust to changes in PAR protein levels. This raises interesting questions on how downstream pathways read out PAR polarity.

Finally, the authors investigate the interplay in robustness between symmetry breaking cues (which similarly are robust to perturbations) and PAR polarity. They find that changes in either system that by themselves are tolerated, sensitize to changes in the other system. E.g., when cortical flow is reduced, changes in PAR dosage that are normally tolerated now result in failure to form polarized PAR domains.

Overall I find all of the experiments very rigorously performed and well documented. The only potential cause for concern for acceptance in the flagship PLOS journal is that there are no mechanistic insights into how downstream pathways robustly translate PAR polarity into asymmetric division. However, this study provides a strong and necessary foundation for such studies.

I have a few smaller points that I think need clarification.

In figure 4 the authors claim that 'When PAR-2 is depleted, the anterior PAR-6 domain expands into the posterior.' Looking at panels C and D, I can't see this. The boundary of the PAR-6 domain seems to stay the same regardless of the levels of PAR-2.

In the experiments probing the spindle positioning pathway, I was expecting to see an analysis of the anterior and posterior pulling forces, a relatively standard approach in the field that directly analyzes forces acting on the spindle rather than relying on secondary readouts like oscillations. Performing this analysis would better inform on whether the robustness of the spindle positioning pathway is at the level of force generation or not. Perhaps the finding that oscillations are not perturbed in par-6(+/-) is due to the relative insensitivity of the assay.

I was hoping for a more detailed discussion placing the findings of this manuscript into the broader field of modeling PAR polarity in the C. elegans embryo. To what extent do current models, including those of the senior author himself, predict that changes in PAR protein level will lead to the observed effects of reducing cortical levels while maintaining overall polarization. The authors write that "Specifically, we predict that polarity outputs in otherwise wild-type embryos should be robust to changes in cue strength and, conversely, that reductions in cue strength should render polarity outputs sensitive to PAR dosage". I puzzled when comparing this with modeling and observations in the 2011 Science publication of the senior author. Quoting from that paper on expectations of the model: "changing the amount of a given PAR protein should result in changes in the steady-state position of the PAR boundary (fig. S6). We found that PAR-2 domains were larger in embryos possessing only one functional par-6 gene copy (Fig. 4, D and E) and in embryos where PAR-2 was overexpressed by optimizing the codon adaptation index (CAI) of the green fluorescent protein (gfp)::par-2 transgene (Fig. 4, F to H)." In the current manuscript, embryos possessing only one par-6 copy do not appear to have an altered PAR boundary. How should I reconcile these seemingly conflicting observations, and what does this say about our current PAR polarity models?

In this sentence, 1D should read 2D: However, embryos heterozygous for par-1, par-3, and pkc-3 did not exhibit such increases (Figure 1D).

Rev. 2:

Rodrigues et al. report the results of a sensitivity analysis examining the effect of reducing PAR protein concentration on worm zygote polarity, asking how downstream pathways function with reduced concentrations. They find a number of phenotypes that are insensitive to changes in Par complex components over some range (that includes the decrease caused by heterozygosity). They first demonstrate that no significant dosage compensation mechanisms are present to account for gene dosage effects, but the amount of PAR on the membrane is affected. They examine zygote asymmetry as a function of PAR levels and find at least some regime where asymmetry is relatively unaffected by changes (not surprisingly at least to the point of the changes caused by heterozygosity). This also holds true for several downstream "outputs", although if the system is sensitized (such as by myosin depletion), it appears the threshold of sensitivity is shifted. Overall the study seems well-executed and the results are clearly communicated and interesting. However, I found the paper to be very confusing and I hope the authors will consider trying to make the paper more accessible before publication with the following suggestions.

- The author's argue that their data "reveal nonlinear relationships between signal inputs and output responses". Not surprisingly, this conclusion is highly dependent on what one calls an input and output. The authors use PAR concentration as a proxy for PAR output but how do they know that the PAR signal that influences the phenotypes they measure is directly correlated to PAR concentration? For example, aPKC catalytic activity could be the relevant output and might not follow simply from its concentration. The assumption that PAR output and concentration are directly related should be made clear to readers, along with any possible limitations.

- Below "0.25" normalized Par-2 it appears that there is no detectable cortical Par-2 (Fig 4H). How is Par-2 polarized at these levels (Figure 4I)?

- The discussion states "One common explanation for robustness to dosage changes is that enzymes are simply present in excess." Why is this "common explanation" introduced so late in the paper and isn't it a possible explanation for the observations?

- "We reasoned that robustness of developmental processes to changes in gene or protein dosage in a given molecular network could arise from:" Related to the previous point, this enumeration leads the reader to believe that this problem has never been considered before in the literature. But there is an extensive literature, both experimental and theoretical, on this problem that is at least partially touched on in the discussion that would be usefully discussed while motivating possible mechanisms.

- I find "quantitative decoupling" to be unnecessarily confusing. Is there a meaningful distinction between quantitative decoupled and the simple and straightforward term "insensitive"?

- "Thus, models generally predict that polarization and asymmetric division of the zygote would be sensitive to PAR dosage changes." The authors should be more explicit by what they mean by "sensitive" in these arguments. It's known that some amount of these factors are required for polarity and asymmetric division so at some level they must be sensitive, but do the models insist that there are no regions that are insensitive, as the authors seem to be implying?

- "Current models for PAR-dependent asymmetric division invoke local concentrations of PAR proteins as the key signals regulating division asymmetry pathways (Galli et al., 2011; Griffin et al., 2011). Yet the phenotypic endpoints of asymmetric division remain wild-type despite significant changes in local PAR concentration. How can we square these observations? " Related to the previous point, I found these statements confusing for two reasons. First, models aren't observations. Second, the question seems like a straw man unless the authors more clearly articulate how the models predict polarity will change with respect to Par concentration.

- "network adaptation - features of the network compensate for imbalance to maintain stable polarity signals, such as changes in feedback strength or pattern; " I was confused as to what the "such as" refers to

- "asymmetric division modules acts to suppress the accumulation of error" The authors should be more specific as to what they mean by "error" and why this error would intrinsically accumulate as they imply (if the error didn't intrinsically accumulate, no action would be required).

---

## [Decision Letter · Decision Letter 2]

19 Sep 2024

Dear Dr Goehring,

Thank you for your patience while we considered your revised manuscript entitled "Nonlinear readout of spatial cues underlies robustness of asymmetric cell division" for publication as a Research Article at PLOS Biology. This revised version of your manuscript has been evaluated by the PLOS Biology editors, the Academic Editor and the two original reviewers.

Based on the reviews, we are likely to accept this manuscript for publication, provided you address the remaining points raised by Reviewer 2 - we do think that the readibility of the paper will be improved, so we do encourage you to consider the changes suggested. Please also make sure to address the data and other policy-related requests stated below.

In addition, we would like you to consider a suggestion to improve the title:

"Asymmetric cell division pathways dependent on PAR polarity are robust to changes in protein dosage and symmetry-breaking cues in C. elegans"

We expect to receive your revised manuscript within two weeks. 

*Published Peer Review History*

*Press*

Sincerely,

Ines

--

Ines Alvarez-Garcia, PhD

Senior Editor

PLOS Biology

Fig. 2B-J; Fig. 3A-H; Fig. 4B-E; Fig. 5B-E; Fig. 6A, B, D, E, G, H; Fig. 7B-F; Fig. 8B, C, E, H, J; Fig. S1B-F and Fig. S2A-F, H

Please also ensure that figure legends in your manuscript include information ON WHERE THE UNDERLYING DATA CAN BE FOUND, and ensure your supplemental data file/s has a legend.

CODE POLICY

DATA NOT SHOWN?

- Please note that per journal policy, we do not allow the mention of "data not shown", "personal communication", "manuscript in preparation" or other references to data that is not publicly available or contained within this manuscript. Please either remove mention of these data in the figure legend of Fig. 2 or provide figures presenting the results and the data underlying the figure(s).

Reviewers' comments

Rev. 1:

I am happy with the changes made to the manuscript in the revision. The inclusion of the spindle severing experiment confirms the results obtained by examining oscillations, and the stability of the PAR-6 boundary upon PAR-2 loss is now clearly explained. I also appreciate the additional discussion of the results in light of previous models. Hence, all my comments have been addressed.

Rev. 2:

My primary criticism of the initial version of the paper was that it was confusing. I appreciate that the authors attempted to revise the paper to make it easier to understand but unfortunately I don't find the revision to be significantly improved in this regard. A good deal of the problem arises from choices that the authors have made to use phrases like "nonlinear readout", "intrinsic robustness", and "quantitative decoupling" (and many more). I specifically noted the latter phrase in my initial review and the authors replied that "The term "quantitative decoupling" was taken from the literature (Kitano 2004)" however I don't believe that they are correct. I didn't find the phrase "quantitative decoupling" in Kitano's robustness review or anywhere else in the literature. In fact, as far as I can tell, there is no difference between that author's usage of "quantitative decoupling" and Kitano's use of "decoupling", with the latter being significantly more straightforward to understand. With that said, I made the same point in my initial review and the authors have revised their paper in the manner that they saw fit so I don't see any reason to delay publication of the paper any further.

---

## [Editor Report · Decision Letter 3]

30 Oct 2024

Dear Dr Goehring,

Thank you for the submission of your revised Research Article entitled "Quantitative perturbation-phenotype maps reveal nonlinear responses underlying robustness of PAR-dependent asymmetric cell division" for publication in PLOS Biology. On behalf of my colleagues and the Academic Editor, François Schweisguth, I am delighted to let you know that we can in principle accept your manuscript for publication, provided you address any remaining formatting and reporting issues. These will be detailed in an email you should receive within 2-3 business days from our colleagues in the journal operations team; no action is required from you until then. Please note that we will not be able to formally accept your manuscript and schedule it for publication until you have completed any requested changes.

PRESS

Sincerely, 

Ines

--

Ines Alvarez-Garcia, PhD

Senior Editor

PLOS Biology
